# Precise control of neural activity using dynamically optimized electrical stimulation

**Nishal Pradeepbhai Shah**[1,2,3]\*[†], **AJ Phillips**[1,3]\*[†], **Sasidhar Madugula**[2,3], **Amrith Lotlikar**[1], **Alex R Gogliettino**[3,4], **Madeline Rose Hays**[3,5], **Lauren Grosberg**[2,3], **Jeff Brown**[1], **Aditya Dusi**[1], **Pulkit Tandon**[1], **Pawel Hottowy**[6], **Wladyslaw Dabrowski**[6], **Alexander Sher**[7], **Alan M Litke**[7], **Subhasish Mitra**[1], **EJ Chichilnisky**[2,3,8]

[1]Department of Electrical Engineering, Stanford, United States; [2]Department of Neurosurgery, Stanford, United States; [3]Hansen Experimental Physics Laboratory, Stanford University, Stanford, United States; [4]Neurosciences PhD Program, Stanford, United States; [5]Department of Bioengineering, Stanford, United States; [6]AGH University of Science and Technology, Faculty of Physics and Applied Computer Science, Krakow, Poland; [7]Santa Cruz Institute for Particle Physics, University of California, Santa Cruz, CA, Santa Cruz, United States; [8]Department of Ophthalmology, Stanford, United States

**\*For correspondence:**
bhaishahster@gmail.com (NPS);
andrewjp@stanford.edu (AJP)

[†]These authors contributed equally to this work

**Competing interest:** The authors declare that no competing interests exist.

**Abstract** Neural implants have the potential to restore lost sensory function by electrically evoking the complex naturalistic activity patterns of neural populations. However, it can be difficult to predict and control evoked neural responses to simultaneous multi-electrode stimulation due to nonlinearity of the responses. We present a solution to this problem and demonstrate its utility in the context of a bidirectional retinal implant for restoring vision. A dynamically optimized stimulation approach encodes incoming visual stimuli into a rapid, greedily chosen, temporally dithered and spatially multiplexed sequence of simple stimulation patterns. Stimuli are selected to optimize the reconstruction of the visual stimulus from the evoked responses. Temporal dithering exploits the slow time scales of downstream neural processing, and spatial multiplexing exploits the independence of responses generated by distant electrodes. The approach was evaluated using an experimental laboratory prototype of a retinal implant: large-scale, high-resolution multi-electrode stimulation and recording of macaque and rat retinal ganglion cells ex vivo. The dynamically optimized stimulation approach substantially enhanced performance compared to existing approaches based on static mapping between visual stimulus intensity and current amplitude. The modular framework enabled parallel extensions to naturalistic viewing conditions, incorporation of perceptual similarity measures, and efficient implementation for an implantable device. A direct closed-loop test of the approach supported its potential use in vision restoration.

## Editor's evaluation

This valuable study proposes a new algorithm for determining the electrical stimulation delivered through a sensory-neural/retinal implant with the aim of improving the perceptual benefit to implant users. The evidence supporting the conclusions is solid, with additional experiments and analyses submitted during the revision having significantly strengthened the study. The work will be of interest to both neuroscientists and neuroengineers.

## Introduction

A major goal of sensory neuroscience is to leverage our understanding of neural circuits to develop implantable devices that can artificially control neural activity for restoring senses such as vision (*Humayun et al., 2012*; *Stingl et al., 2013*; *Palanker et al., 2020*; *Beauchamp et al., 2020*; *Chen et al., 2020*), audition (*Gaylor et al., 2013*), and somatosensation (*Johnson et al., 2013*; *Flesher et al., 2016*; *Salas et al., 2018*). Recent innovations in large-scale and high-resolution electrical stimulation hardware hold great promise for such applications. However, the effect of stimulation with many closely spaced electrodes on neural activity is generally complex and nonlinear, which severely limits the ability to produce the desired spatiotemporal patterns of neural activity for sensory restoration.

This paper presents a novel approach to this problem in the context of an epiretinal implant for restoring vision in people blinded by photoreceptor degeneration (*Humayun et al., 2012*; *Beyeler et al., 2019*; *Bloch, 2020*). Epiretinal implants electrically activate retinal ganglion cells (RGCs) that have survived degeneration, causing them to send artificial visual signals to the brain. After initial successes, progress toward restoring high-fidelity natural vision using this approach has slowed, likely in part due to indiscriminate activation of many RGCs of different cell types and a resulting inaccurate neural representation of the target stimulus. One reason for this indiscriminate activation is the difficulty of predicting the neural activity evoked by multi-electrode stimulation based on the activity evoked by single-electrode stimulation.

Here, we present a data-driven optimization approach to bypass this problem by dynamically combining simpler stimulation patterns (*Choi et al., 2016*; *Beauchamp et al., 2020*; *Tafazoli et al., 2020*; *Haji Ghaffari et al., 2021*; *Vasireddy et al., 2023*) using temporal dithering and spatial multiplexing. The presented solution is divided into three steps, allowing it to be modified for a wide range of neural systems and implants. First, we develop a simple, explicit model of how the visual image can be reconstructed from the activity of many RGCs of diverse types accessed by the multi-electrode array. Second, we avoid the complexity of nonlinear electrical stimulation by empirically calibrating RGC responses to a collection of simple single-electrode stimuli which can then be combined asynchronously and sparsely to reproduce patterns of neural activity. Finally, we optimize visual scene reconstruction by greedily selecting a sequence of these simple stimuli, temporally dithered to exploit the high speed of electrically evoked neural responses, and spatially multiplexed to avoid interactions between nearby electrodes. These three steps result in a dynamically optimized stimulation paradigm: the visual stimulus is transformed into a spatiotemporal pattern of electrical stimuli designed to produce a pattern of neural activity that is most effective for vision restoration given the measured limitations of the neural interface.

This dynamic optimization approach was tested and evaluated using large-scale multi-electrode stimulation and recording ex vivo from the macaque and rat retinas, a lab prototype for a future implantable system. The method produced substantial improvements in stimulus reconstruction compared to existing methods, by appropriately activating ON and OFF RGCs over space. The algorithm was useful in identifying a subset of the most effective electrodes for a particular retina, which could substantially reduce power consumption in an implant. Extensions of the algorithm can in principle be used to translate the approach to naturalistic viewing conditions with eye movements, and to exploit perceptual metrics to further enhance the quality of reconstruction.

## Results

First, we frame the translation of a visual stimulus into electrical stimulation as an optimization problem and present a greedy temporal dithering algorithm to solve it efficiently. Then, we use the ex vivo lab prototype to evaluate the performance of the approach. We compare it with existing methods and develop extensions for spatial multiplexing, natural viewing with eye movements and perceptual quality measures.

The ex vivo lab prototype consists of electrical recording and stimulation of the macaque and rat retinas with a large-scale high-density multi-electrode array (512 electrodes, 60 μm spacing). The visual and electrical response properties of all recorded cells are estimated by direct measurements, using experimental methods described previously (*Field and Chichilnisky, 2007*; *Jepson et al., 2013*; *Grosberg et al., 2017*) (see Methods). These data provide reliable experimental access to complete

populations of ON and OFF parasol cell types in macaque retina, so these two cell types are the focus of the empirical analysis.

## Dynamic optimization to approximately replicate neural code

Converting a visual stimulus into effective electrical stimulation can be framed as an optimization problem. Using the terminology of optimization, the three key components are the *objective function*, the *constraints*, and the *algorithm* (*Figure 1A*). The *objective* function to be minimized is identified as the difference between the target visual stimulus and a reconstruction of the stimulus from the neural responses, as a proxy for how the brain could use the signal for visual inference. However, certain *constraints* are imposed by electrical stimulation, which provides imperfect control over the activity of a population of cells. Hence, the optimization *algorithm* must convert incoming visual stimuli into electrical stimuli, such that the stimulus reconstructed from electrically evoked responses matches the true stimulus as closely as possible.

### Objective: reconstructing the visual stimulus from neural responses

The *objective* of electrical stimulation in this context is to reproduce, as closely as possible, a visual sensation that would be produced by normal light-evoked responses. However, it is not known how the brain interprets RGC light responses, and thus how to frame the problem computationally. As a simple proxy, the objective is defined by reconstructing the visual image as accurately as possible from evoked RGC spikes (see Discussion) and then evaluating the difference between the reconstruction and the original image. For simplicity, linear reconstruction is assumed, and the objective is the minimum squared error between the target image and the reconstructed image.

Specifically, for a target image shown to the retina in the experimental lab prototype setting, the reconstructed stimulus is modeled as the linear superposition of spatial filters, each associated with a particular RGC, weighted by the corresponding RGC response (*Figure 1B*; *Warland et al., 1997*; *Brackbill et al., 2020*). The optimal linear reconstruction filter for each ON and OFF parasol cell was approximated using the measured spatial receptive field of the cell obtained with white noise stimulation, scaled to predict the average spike count recorded within 50 ms of the onset of a flashed checkerboard stimulus. Note that in an implanted blind retina, the reconstruction filter for each cell would have to be estimated in a different way (*Zaidi et al., 2022*, see Discussion).

### Constraint: calibrating the collection of neural responses that can be electrically evoked

The limited precision of electrical stimulation *constrains* our ability to produce desired response patterns in the RGC population. To optimize stimulation under this constraint, it would be ideal to have a model that characterizes how RGCs respond to arbitrary electrical stimulus patterns produced with the electrode array. Unfortunately, estimating this model is difficult due to nonlinear interactions in neural activation resulting from current passed simultaneously through multiple electrodes (*Jepson et al., 2014*).

An alternative is to use a limited *dictionary* of responses evoked by simple current patterns. This dictionary was *calibrated* empirically in advance (*Figure 1C*). Specifically, current was passed through each of the 512 electrodes individually at each of 40 current levels (logarithmically spaced over the range 0.1–4 μA) and the response probability for each recorded cell was estimated using the fraction of trials in which an evoked spike was recorded from the cell. The evoked spikes were identified using a custom spike sorting algorithm that estimated the electrical artifact produced by stimulation and matched the residual recorded voltage to template waveforms of cells previously identified during visual stimulation (*Mena et al., 2017*). In general, even though some electrical stimuli selectively activated one cell, many stimuli simultaneously activated two or more cells, often due to axonal stimulation (*Figure 1C*; *Grosberg et al., 2017*). Also, high-amplitude stimuli tended to evoke spikes in cells with receptive fields off the electrode array via their axons. These cases were detected by identifying bidirectional spike propagation to the edge of the electrode array and were removed from the dictionary (*Grosberg et al., 2017*; *Tandon et al., 2021*).

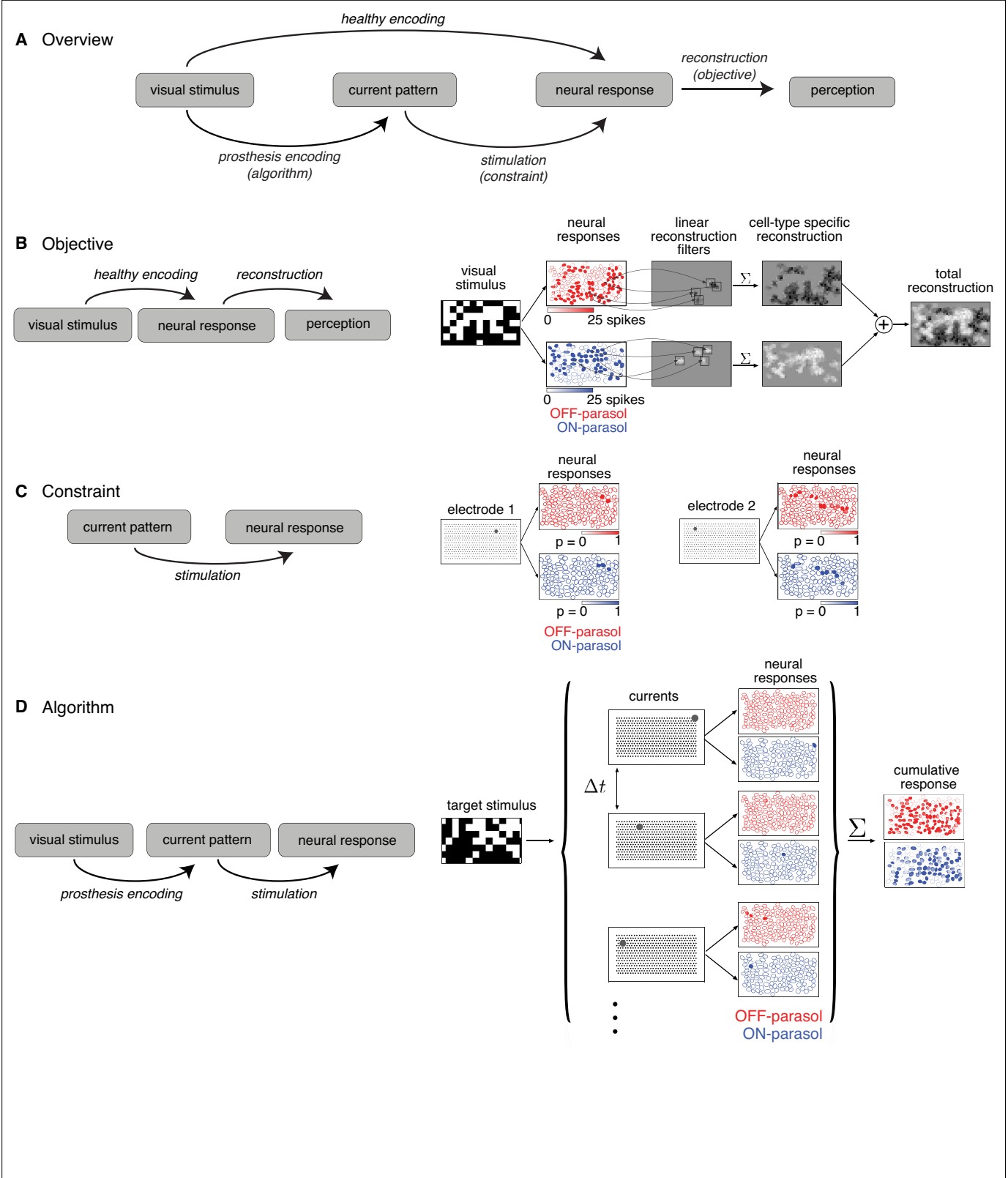

**Figure 1.** Algorithmic components of the proposed framework for electrical stimulation. (**A**) In a healthy retina, the visual stimulus is encoded in the neural response pattern of retinal ganglion cells (RGCs; top row). In a retina with an implant, the visual stimulus is encoded into current patterns, which generate neural response patterns (bottom row). In either case, the neural responses are eventually processed by the brain to elicit perception, through a process assumed to involve reconstruction of the image. Selecting the appropriate electrical stimulation can be framed as an optimization problem,

*Figure 1 continued on next page*

*Figure 1 continued*

in which the goal is to identify an *algorithm* (prosthesis encoding) that achieves an *objective* (reconstruction error) while operating under *constraints* (electrical stimulation). (**B**) *Objective*: Linear reconstruction of visual stimulus by summing cell-specific spatial filters, weighted by spike counts. Receptive fields of ON (blue) and OFF (red) parasol cells in a population are shown. (**C**) *Constraint*: Characterizing electrically evoked RGC responses with a dictionary of stimulation patterns. Example dictionary elements, with cells shaded according to evoked response probability. A single-electrode stimulated multiple cells, indicating poor selectivity. (**D**) *Algorithm*: Run-time usage of the artificial retina. Exploiting the slow visual integration time, distant electrodes are stimulated in fast sequence. The resulting neural response is the summation of spikes elicited in each time step.

## Algorithm: greedy temporal dithering to approximate the optimal spatio-temporal electrical stimulus

Because a single-electrode stimulus generally cannot create a pattern of activity across the RGC population that accurately encodes a visual image, multiple stimuli must be combined, while also avoiding the nonlinear interactions mentioned above. This was achieved by rapid interleaving or *temporal dithering* of a diverse collection of single-electrode stimuli. The effectiveness of this method relies on assuming that if many such stimuli are provided in rapid succession (e.g. 0.1-ms interval), they evoke visual sensations similar to those that would be produced by simultaneous stimulation (*Figure 1D*), because of long visual integration times in the brain (e.g. tens of ms, see Discussion).

Under this assumption, the optimization problem reduces to finding a sequence of dictionary elements $\{c_1, \cdots, c_T\}$ that minimizes the expected squared error between the target visual stimulus and reconstructed responses based on the total spike count:

$$c_1, ..., c_T = argmin\, E_{r_i \sim Bernoulli\,(p_{c_i})}\, \|s - A\,(r_1 + ... + r_T)\,\|^2 \tag{1}$$

where $s \in (pixels)$ is the target visual stimulus, $A \in (pixels \times cells)$ is the stimulus reconstruction filter, and $r_i \in (cells)$ is a vector of the spike counts in the population of cells generated by stimulation $c_i$, with spikes being drawn according to Bernoulli processes with probabilities $p_{c_i} \in (cells)$.

In order to create an effective stimulation sequence, a straightforward, real-time method is to greedily select a stimulus at each step that minimizes the predicted error between the reconstruction and the target image. Although the greedy approach is not necessarily optimal (as explained later), it does allow for effective real-time implementation.

A crucial assumption of the algorithm is that the total expected spike count evoked by a sequence of electrical stimuli is the sum of the expected spikes for all the individual stimuli. However, when a cell is repeatedly stimulated, the activation probabilities associated with later stimuli are reduced, because of biophysical refractoriness. To avoid this non-independence, the stimulus at each time step is chosen from a 'valid' subset of the dictionary that does not include cells that were targeted recently (see Methods).

## Greedy temporal dithering outperforms existing static methods

The greedy temporal dithering algorithm was evaluated on data collected using the laboratory experimental prototype for a retinal implant. After calibrating the responses of all recorded RGCs to all available single-electrode stimuli, the greedy temporal dithering stimulation sequence corresponding to a specific visual target (usually, a random checkerboard image) was calculated, as described above (*Equation 1*). Then, the visual target was linearly reconstructed from the stimulation sequence using samples drawn from the single-electrode calibration data, under the assumption that temporal dithering maintains the independence of the responses evoked by single-electrode stimuli. Note that this assumption was later tested (see below).

The results of the analysis obtained by sampling from calibration data reveal the inferred visual reconstruction that is possible with greedy temporal dithering. During the stimulation sequence, the reconstructed image slowly built up to a spatially smooth version of the target image (*Figure 2*). Not surprisingly, ON (OFF) parasol cells were stimulated more than OFF (ON) parasol cells in bright (dark) regions of the target stimulus. Moreover, the reconstruction for individual trials was similar to the average across multiple trials, indicating that the noise from inter-trial response variation was relatively small.

To quantify the performance of the greedy temporal dithering algorithm, its reconstruction error was compared to the error of an optimized approach for existing retinal implants, which map the

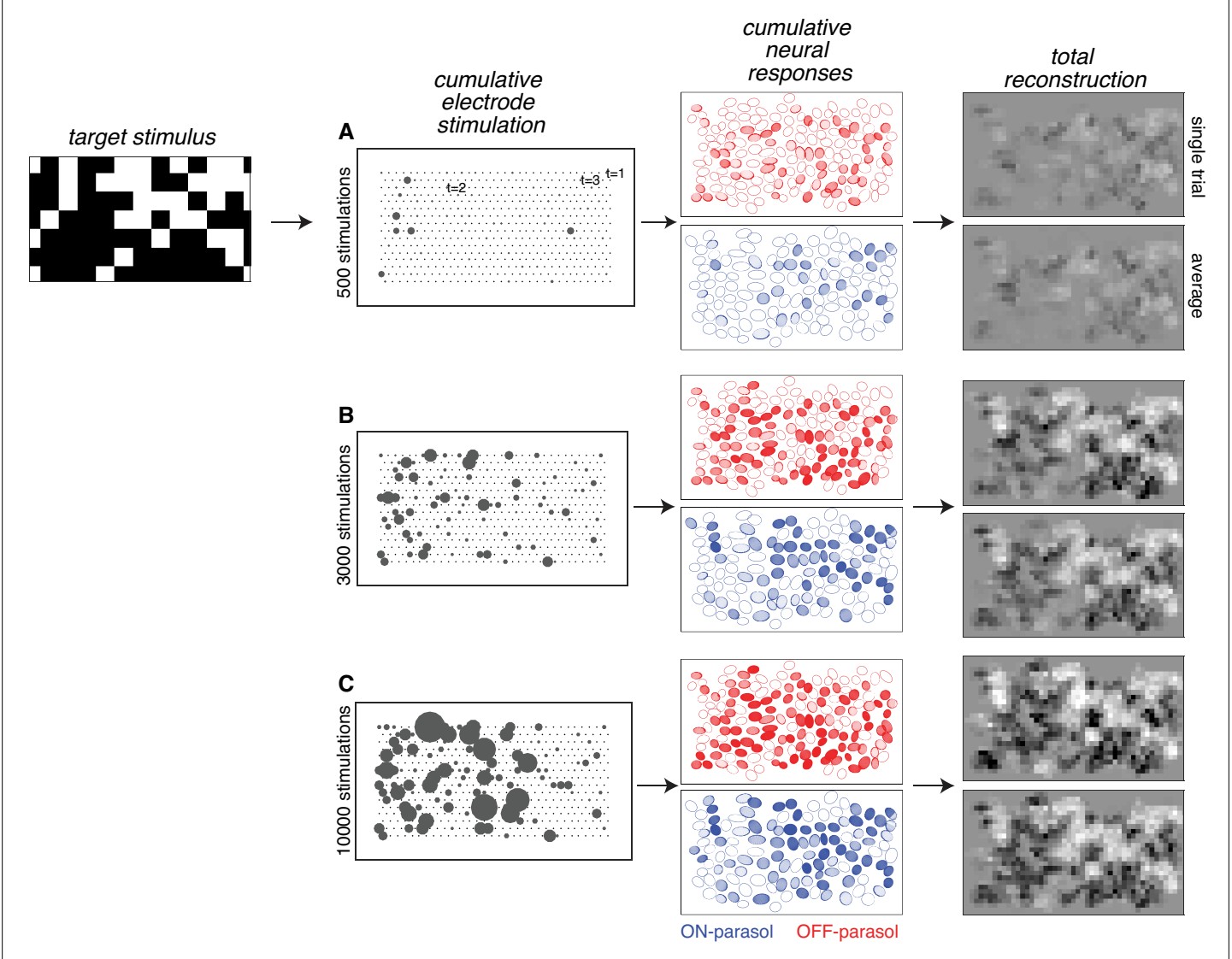

**Figure 2.** Visual stimulus reconstruction achieved using the greedy temporal dithering algorithm. White noise target image shown on left. First column: cumulative stimulation count across electrodes after 500, 3000, and 10,000 electrical stimuli (A, B, and C, respectively). Second column: responses for ON (blue) and OFF (red) parasol cells, sampled according to the single-electrode calibration data. Shade indicates the cumulative number of spikes. Third column: single-trial and trial-averaged reconstruction of the target stimulus.

intensity of the visual stimulus near each electrode to its stimulation current amplitude or frequency (*Humayun et al., 2012*; *Stingl et al., 2013*; *Palanker et al., 2020*). The performance of this static pixel-wise mapping was simulated with the lab prototype. Specifically, the current passed through each electrode was determined by a sigmoidal function of the intensity of the visual stimulus near the electrode, optimized at each electrode to minimize the reconstruction error across a training set of random checkerboard images (see Methods). This approach provides a generous upper bound to the performance of existing implants, because it uses actual response probabilities to optimize each sigmoidal function rather than relying on much more limited patient feedback, as is the case in existing retinal implants. Even using this generous upper bound, static pixel-wise mapping resulted in significantly less accurate reconstruction of the target image (*Figure 3D, H*), likely due to coactivation of overlapping ON and OFF cell types.

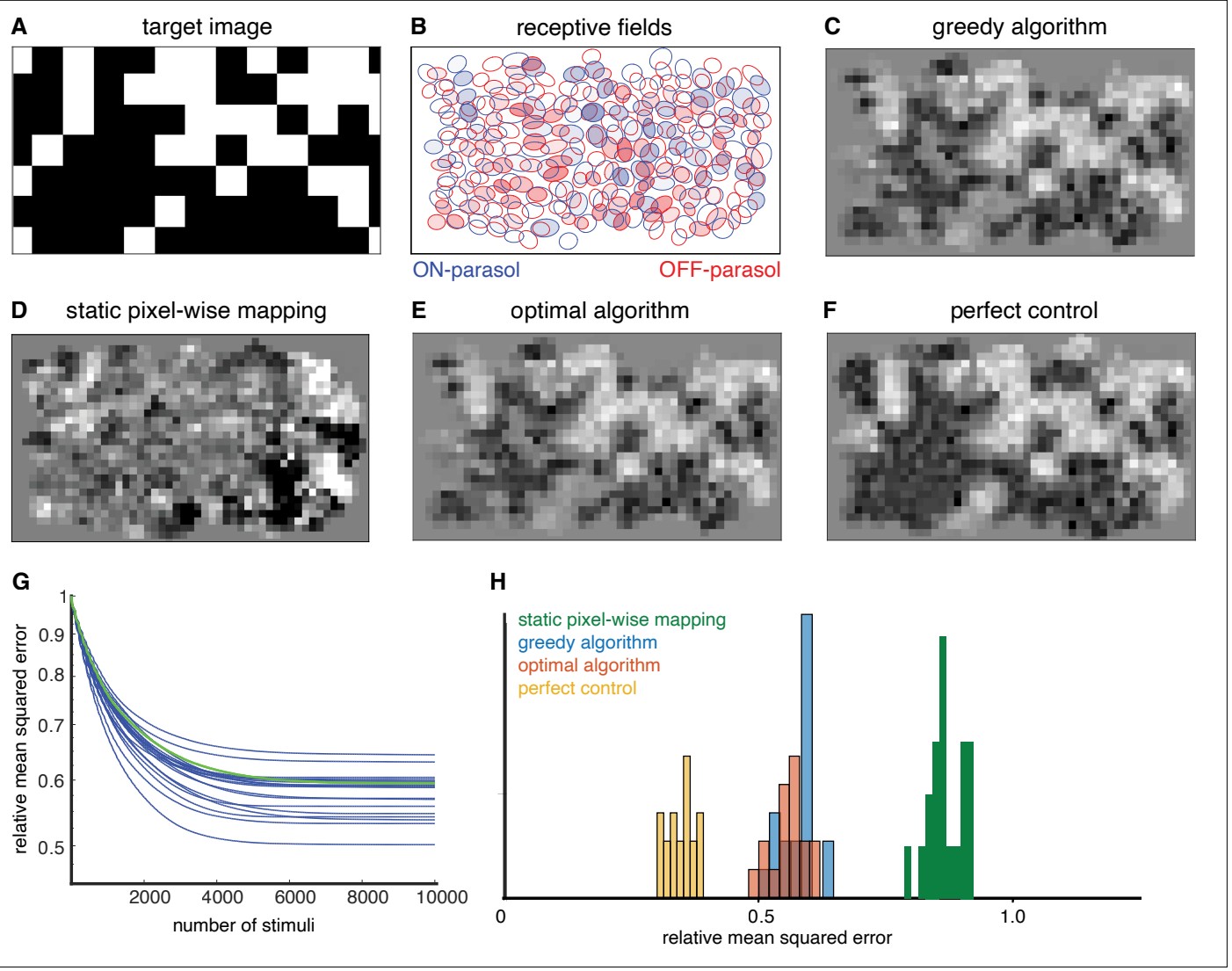

**Figure 3.** Quantifying the performance of dynamically optimized stimulation. (**A**) A sample target checkerboard image. (**B**) ON and OFF receptive fields shaded with the expected summed response from greedy temporal dithering. Achieved reconstructions are shown for (**C**) greedy temporal dithering using calibrated responses to single-electrode stimulation (8448 electrical stimuli), (**D**) static pixel-wise mapping approximating existing open-loop systems (5023 stimuli), (**E**) a lower error bound on the optimal algorithm for a single-electrode dictionary (1850 stimuli), and (**F**) perfect control with the available reconstruction filters. (**G**) Reconstruction error (relative mean squared error) between target and the expected achieved perception for 20 different targets (blue lines), with the example from C indicated with the green line. (**H**) Histogram of relative performance of the above approaches across 20 target images.

## Greedy temporal dithering is nearly optimal given the interface constraints

What factors could improve the performance of the dynamically optimized stimulation approach? Broadly, performance could be limited either by the algorithm (greedy selection of electrical stimuli) or by the constraints of the interface (limited control of neural activity afforded by single-electrode stimulation).

To test whether performance could be improved with a different algorithm, the greedy approach was compared with a nearly optimal algorithm. The original optimization problem can be reformulated as:

$$minimize_{w \geq 0} \parallel s - ADw \parallel^2 + v^T w \; such \; that \; w \in Z_+$$

where $s \in (pixels)$ is the target stimulus, $A \in (pixels \times cells)$ is the reconstruction filter, $D \in (cells \times stimuli)$ is a matrix of all response probabilities in the dictionary, $v \in (stimuli)$ is the variance associated with dictionary elements and $w \in (stimuli) \geq 0$ indicates the number of times each dictionary element is used. Because $w$ is an integer, this optimization problem is difficult to solve. However, an upper bound on the performance gap between the greedy algorithm and the optimal algorithm can be obtained by allowing non-integer values of $w$. Across multiple target images, the gap was low (<10%, 'optimal algorithm' in *Figure 3E, H*), suggesting that the approximate nature of the greedy algorithm is not a substantial source of error in the present conditions.

To test whether performance could be improved with a more precise neural interface, the reconstruction error was compared with perfect control, in which any desired response pattern can be produced. The performance with perfect control was estimated by the solving the following optimization problem:

$$minimize_{r \geq 0} \; \|s - Ar\|^2 \; such \, that \; r \in Z_+$$

where $r \in (cells)$ is the vector of spike counts. This optimization problem was solved by relaxing the integer constraint on $r$ to obtain an upper bound on the performance gap between a single-electrode dictionary and an ideal dictionary. Across multiple targets, this gap was substantial (>40%, 'perfect control' in *Figure 3F, H*).

Thus, although the greedy algorithm is nearly optimal for a single-electrode dictionary, the reconstruction performance of an artificial retina could be improved by enhancing the dictionary (e.g. by using calibrated, optimized multi-electrode stimulation patterns; *Jepson et al., 2014*; *Vasireddy et al., 2023*).

## Closed-loop experimental validation of greedy temporal dithering

The performance of the greedy temporal dithering algorithm was next tested empirically using closed-loop recording and stimulation in the isolated rat retina (see Methods). We compared the

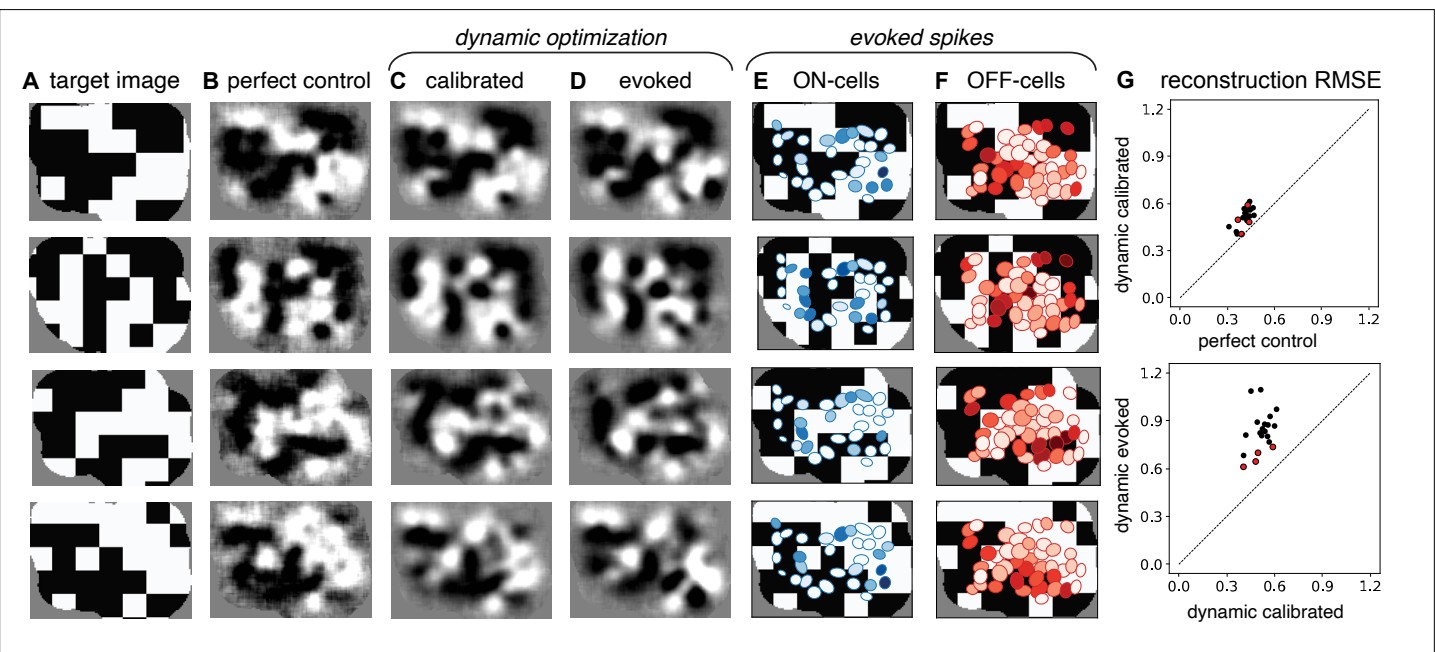

**Figure 4.** Experimental validation of dynamically optimized stimulation in the rat retina. (**A**) Four sample target checkerboard images. Achieved reconstructions for these images are shown (**B**) assuming perfect control of retinal ganglion cell (RGC) firing with the available reconstruction filters, (**C**) using greedy temporal dithering based on calibrated single-electrode responses, and (**D**) and using the RGC responses evoked during electrical stimulation with greedy temporal dithering. (**E, F**) ON and OFF receptive fields shaded with the total number of evoked spikes. (**G**) Reconstruction error (relative mean squared error) across 20 target images for perfect control vs. greedy temporal dithering using calibrated responses (top) and for greedy temporal dithering using calibrated responses vs. evoked responses (bottom). Red points correspond to the four targets shown. Evoked RGC responses are averaged over 25 trials for each target (**D–G**).

reconstruction of the visual stimulus from RGC responses evoked by the stimulation sequence to the reconstruction obtained using samples drawn from the single-electrode calibration data. First, reconstruction filters were obtained using visual stimulation and recording, and the responses to single-electrode stimulation were calibrated as described above. Then, the greedy temporal dithering stimulation sequence was computed during the experiment and delivered to the retina at an expanded 3-ms stimulation interval to facilitate spike sorting (see Methods). The evoked RGC responses to this sequence were then recorded and analyzed. The reconstructions obtained with the evoked RGC responses captured much of the spatial structure in each target image (*Figure 4A–D*). Notably, the spatial structure of reconstructions using the calibrated responses to single-electrode stimulation ('calibrated' in *Figure 4*, similar to *Figures 2 and 3*) and the RGC responses evoked by the stimulation sequence ('evoked' in *Figure 4*) were similar. These reconstructions reached asymptotic quality with 225 ± 29 stimulations (not shown), a significantly smaller number than in tests obtained with macaque retina (*Figure 3G*), likely in part because of the smaller number of cells in the rat recordings (see Discussion). Importantly, the reconstruction performance benefited from differential activation of ON and OFF cells over space in a way that reflected the spatial distribution of intensities in each target image (*Figure 4E, F*). This observation highlights the importance of electrical stimulation which approximates naturalistic RGC responses, in comparison with the static pixel-wise approaches used in existing implants.

The limitations to the performance of the dynamically optimized stimulation were further explored with two comparisons. First, the reconstruction error obtained using calibrated responses to single-electrode stimulation was higher than the error obtained under an assumption of 'perfect control' (i.e. that all recorded cells can be activated independently), reflecting the limitations of single-electrode stimulation (*Figure 4G*, top), as shown earlier (*Figure 3H*). Second, the reconstruction error obtained using the RGC responses evoked by the stimulation sequence was higher than the error obtained using calibrated responses to single-electrode stimulation (*Figure 4G*, bottom), presumably reflecting non-stationarity in the ex vivo recordings and/or failures of independence due to temporal dithering (*Figure 4G*, see Discussion).

## Spatial multiplexing increases throughput within the visual integration window

The main requirement of temporal dithering – the independence of responses generated by individual electrical stimuli, and their summation within a time window for visual perception – could limit the throughput of electrical stimulation. Although independence can be ensured by spacing single-electrode stimuli widely in time, this approach could make it difficult to deliver many electrical stimuli within a visual integration window (e.g. tens of ms). One approach to maximizing the number of stimuli that can be delivered is *spatial multiplexing*, in which multiple electrodes are used simultaneously for stimulation if they are known to affect the firing probabilities of disjoint sets of cells. A simple example

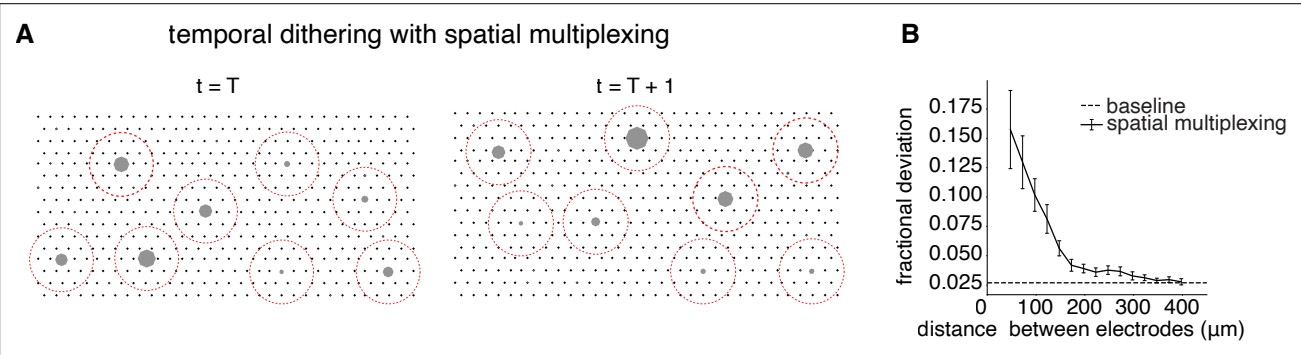

**Figure 5.** Spatial multiplexing by simultaneous stimulation of distant electrodes. (**A**) Visualization of temporally dithered and spatially multiplexed stimulation. At each time step, multiple single-electrode stimuli are chosen greedily (gray circles) across the electrode array (black dots), separated by a spatial exclusion radius (red circles). (**B**) Estimation of the spatial exclusion radius using 754 total electrode pairs across 7 parasol cells from 4 peripheral macaque retina preparations (see Methods). Interaction between electrodes is measured by fractional deviation in activation threshold for a given cell on a primary electrode (ordinate) resulting from simultaneous stimulation of another electrode with identical current amplitude at varying separations (abscissa). Baseline represents the variability associated with estimating single-electrode activation thresholds.

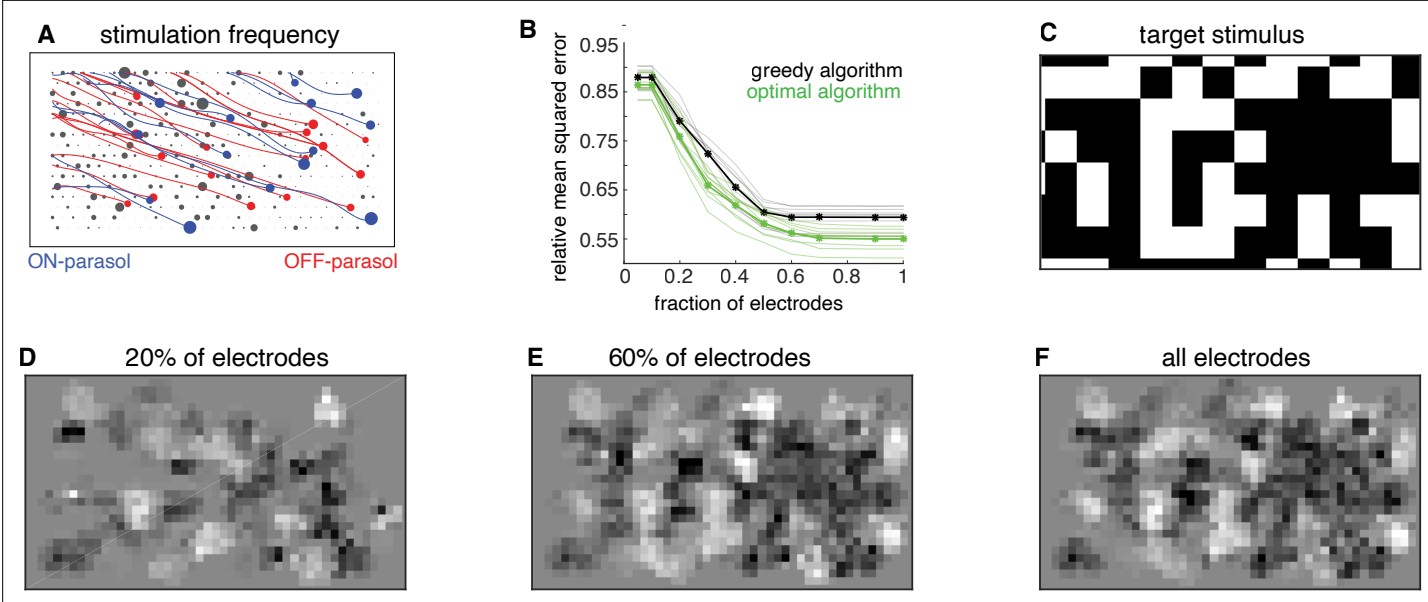

**Figure 6.** Subsampling electrodes for hardware efficiency. (**A**) Frequency of stimulating different electrodes (size of gray circles), overlaid with axons (lines), and somas (colored circles) inferred from spatiotemporal spike waveform across the electrode array recorded from each cell. (**B**) Reconstruction error as a function of the fraction of electrodes included in the dictionary (black, thin lines correspond to different target images) and average over 20 target images (black, thick line). Different collections of target stimuli were used for electrode selection and reconstruction performance evaluation. Lower bound on error of any algorithm for the subsampled dictionaries for individual targets (green, thin lines) and averaged across targets (green, thick line). (**C**) Example target image. (**D–F**) Reconstructed images using the dictionary with most frequently used 20%, 60%, and 100% of electrodes, respectively.

would be if electrodes separated by more than a particular distance $D$ always influenced the firing of disjoint sets of cells. In this case, implementing a circular *exclusion zone* with radius $D$ around each electrode at each time step of temporal dithering would be expected to produce the same cellular activation as was obtained with calibrated single-electrode stimulation. As in the original temporal dithering approach, at the subsequent time step, electrical stimuli with nonzero activation probability for any recently targeted cell would be omitted from consideration (*Figure 5A*).

To identify a spatial exclusion zone and test its effectiveness, activation curves corresponding to single-electrode stimulation were compared to activation curves obtained with additional simultaneous stimulation using a nearby secondary electrode at the same current level (see Methods). Examination of many electrode pairs over a range of distances and multiple cells (*Figure 5B*) revealed a systematic decrease of the interaction between stimulating electrodes as a function of distance. On average, the activation probability of a cell in response to single-electrode stimulation was affected relatively little (<5% fractional change in threshold) for a secondary electrode 200µm away. Thus, two electrodes more than this distance apart were unlikely to substantially influence the activation probability of the same cell(s). This suggests that spatial multiplexing of stimuli outside a spatial exclusion radius is a practical strategy for high-throughput stimulation with temporal dithering.

## Dynamic optimization framework enables data-driven hardware design

The dynamic optimization framework suggests further optimizations for hardware efficiency. Because the greedy temporal dithering algorithm chooses electrodes in a spatially non-uniform manner over the array (*Figure 6A*), restricting stimulation to a more frequently chosen subset of electrodes could enhance efficiency. To test this possibility, the algorithm was applied with dictionaries restricted to a subset of the most frequently used electrodes, and calibrated performance was evaluated on twenty visual targets. In general, restricted dictionaries would be expected to reduce reconstruction performance. However, a minimal (<5%) increase in reconstruction error was observed if the number of available electrodes was reduced by up to 50% (*Figure 6B–F*). Note that this increase was not due to the greedy nature of stimulation, because a lower bound computed for an optimal algorithm showed similar behavior (*Figure 6B*). These observations suggest a strategy for efficient implant operation in

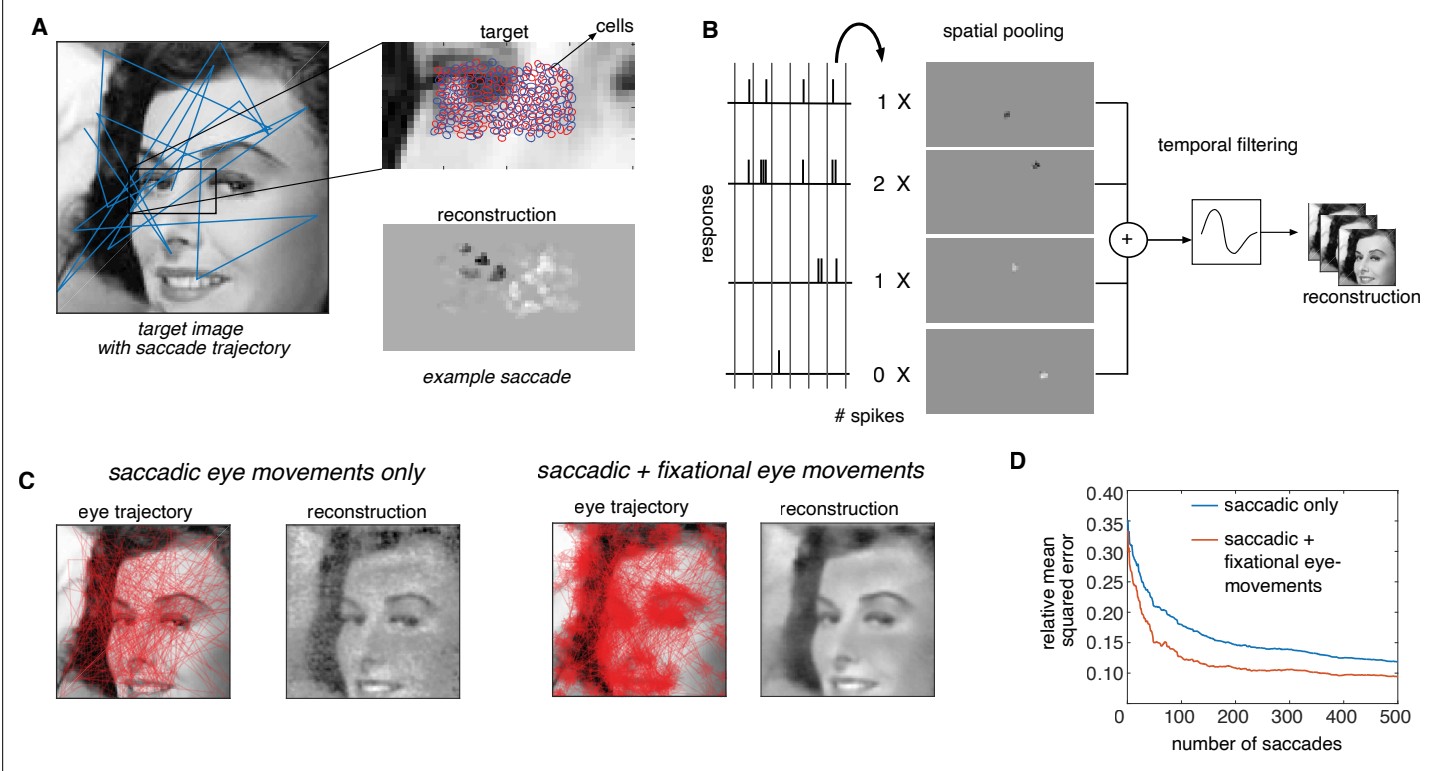

**Figure 7.** Extension of dynamically optimized stimulation to naturalistic conditions with eye movements. (**A**) Conversion of a visual scene into dynamic stimulus. A target visual scene (left), with sample eye movement trajectory (blue). For each eye position, the population of ganglion cells accessible by the implant views a small portion of the visual scene (top right). The reconstructed stimulus for each patch captures the local stimulus information (bottom right). (**B**) Spike trains passed through a spatiotemporal reconstruction filter of the dynamic stimulus video. For simplicity, a rank one filter was used, which spatially filtered each spike bin independently, and then filtered the reconstructed stimulus video in time. (**C**) Final reconstruction performance over a sequence of saccades, in the absence (left) and the presence (right) of small fixational eye movements. (**D**) Reduction in reconstruction error of the visual scene as a function of the number of saccades, in the absence (blue) and the presence (orange) of fixational eye movements.

The online version of this article includes the following video for figure 7:

**Figure 7—video 1.** Greedy temporal dithering and spatial multiplexing in natural viewing conditions.

https://elifesciences.org/articles/83424/figures#fig7video1

a retina-specific manner: identify the most frequently used ~50% of electrodes during calibration and permanently turn off the remaining electrodes during run-time usage. Such a reduction in the set of stimulated electrodes could lead to reduced memory access and power consumption, with little loss in performance. Thus, applying the algorithmic framework to the ex vivo lab prototype leads to insights relevant to the development of an in vivo implant.

## Dynamic optimization framework extends to naturalistic viewing conditions

For practical application, the dynamically optimized stimulation approach must be extended to naturalistic viewing conditions, in which saccadic and fixational eye movements move the fovea over the scene for high-resolution vision (*Figure 7A*). Similarly, an implant fixed on the retina would move over the scene as the eye moves, and would only transmit the information about its restricted view of the image. The dynamic optimization framework extends naturally to this situation.

First, the objective, the constraint, and an algorithm for the corresponding optimization problem are identified. The objective function is modified to minimize the error between the original stimulus video and a video reconstructed from the RGC spike trains. For simplicity, a spatiotemporal reconstruction filter is used with separable spatial and temporal components and the same time course (with opposite polarity) for ON and OFF parasol cells (*Figure 7B*, see Methods). The constraints

(measured electrical stimulation properties) are unchanged. Finally, the algorithm is adapted by choosing a dictionary element for each time step greedily to minimize the average error between the recent frames of the target stimulus seen by the implant and the corresponding frames of the reconstruction (see Methods).

This modified algorithm was evaluated using simulations of naturalistic viewing. For a given scene, a dynamic visual stimulus was generated by simulating saccadic eye movements with random inter-saccade intervals and random fixation locations with a preference for regions of the scene with high spatial-frequency content (*Yarbus, 1967*). Optionally, fixational eye movements were simulated by jittering the visual stimulus with Brownian motion (see Methods). As before, stimulation patterns were determined using greedy temporal dithering, and reconstruction was performed from calibrated responses to single-electrode stimulation. The dynamic visual stimulus covering the collection of recorded cells was reconstructed from the evoked spikes and the full visual scene was then assembled by stitching together the parts of the scene covering the cells at each time step.

The assembled visual scene closely matched the target, capturing many of its fine details (*Figure 7C*, *Figure 7—video 1*). Interestingly, the reconstructed visual scene was smoother and more accurate (lower reconstruction error) when fixational eye movements were simulated along with saccades (*Figure 7C*). Specifically, for the same final reconstruction error, a ~4× reduction in the number of required saccades and hence the number of required electrical stimuli was observed in the presence of fixational eye movements (*Figure 7D*). Hence, the performance of greedy temporal dithering translates to natural viewing conditions and reveals that more accurate image reconstruction is possible with fixational eye movements (*Wu et al., 2024*).

## Optimizing stimulation using a perceptual similarity measure

The framework provides a natural way to use alternative metrics to optimize visual perception evoked by electrical stimulation. Specifically, the mean squared error (MSE) measure of reconstruction accuracy, while convenient, does not accurately capture perceived differences in image content, whereas error metrics such as Structural Similarity (SSIM) more closely parallel perception (*Wang et al., 2004*). To identify a nearly optimal sequence of stimuli with SSIM as the objective, an exhaustive approach was used to optimize across all possible stimulation sequences for every eye location (see Methods). The SSIM and MSE metrics produced similar reconstructions when the number of electrical stimulation patterns was unlimited (*Figure 8B*). This suggests that the choice of reconstruction error metric may not be important for an implant that can stimulate at high rates. However, SSIM optimization produced higher-quality reconstructions when the number of electrical stimuli was limited (*Figure 8C*), a constraint that may be relevant with short visual integration times or low power limits in an implant. Thus, the greedy dithering approach with a perceptually accurate reconstruction metric could lead to

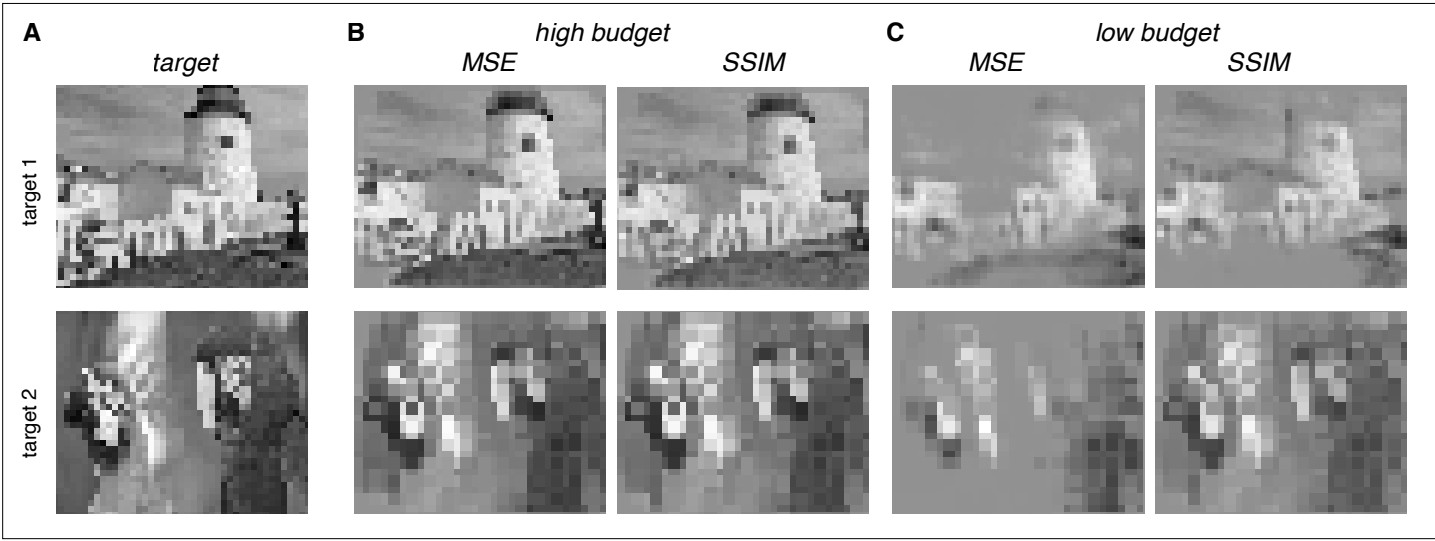

**Figure 8.** Extension of dynamically optimized stimulation using Structural Similarity (SSIM) perceptual error metric. (**A**) Two target images. (**B**) Reconstruction with MSE and SSIM error metrics for greedy temporal dithering, with a high budget. (**C**) Same as B, with a low budget.

higher performance in an implanted device, though additional developments will be needed before such an optimization can be performed in real time.

## Discussion

This paper presents a dynamically optimized electrical stimulation approach to improve the performance of sensory electronic implants. Greedy temporal dithering and spatial multiplexing address the challenges of precisely controlling the activity of diverse cell types in a neural population by rapidly delivering a sequence of simple electrical stimuli with independent effects within a visual integration window. This approach avoids nonlinear interactions resulting from simultaneous multi-electrode stimulation while providing enough flexibility to elicit rich spatiotemporal response patterns using a sequence of single-electrode stimulation patterns. The greedy temporal dithering and spatial multiplexing approach outperforms existing approaches (*Figures 2–5*) and enables efficient neural interface development (*Figure 6*), potentially incorporating naturalistic viewing with eye movements (*Figure 7*) and/or perceptual similarity metrics (*Figure 8*).

The performance of the temporal dithering algorithm was primarily evaluated using calibrated responses from single-electrode stimulation (*Figures 2, 3, and 6–8*). In addition, validation of temporal dithering (*Figure 4*) was performed in closed-loop experiments by delivering the optimized electrical stimulation sequence and analyzing the evoked RGC responses. The reconstructions using these evoked RGC responses captured much of the spatial structure in each target image. However, these reconstructions were less accurate than the reconstructions based on the calibrated responses to single-electrode stimulation (which were used to compute the temporal dithering stimulation sequence during the experiment). At least two factors could contribute to this discrepancy: (1) non-stationarities in the electrical responses of RGCs in the ex vivo retina preparation, due to physiological changes and/or movement, and (2) failures of independence of interleaved stimulation over time relative to isolated single-electrode stimulation. Further experimental work will be needed to distinguish these possibilities. In addition, validation of temporal dithering at shorter stimulation intervals more relevant for in vivo application will require the development of spike sorting approaches that reliably operate in the presence of complex electrical artifacts.

### Assumptions underlying the temporal dithering and spatial multiplexing approach

The presented approach relies on several assumptions regarding how the brain uses RGC responses for vision. A major assumption is that downstream processing of RGC responses is slow, integrating over tens of milliseconds, so that perception depends primarily on the total number of evoked spikes within this time interval. The assumption of slow visual processing is based on evidence ranging from flicker fusion experiments to the time scale of synaptic transfer to neurophysiological tests of temporal integration (*Wandell, 1995*; *Tadin et al., 2010*; *Samaha and Postle, 2015*; *Wutz et al., 2016*; *Borghuis et al., 2019*). However, there is also empirical evidence that the temporal precision of spikes in RGCs in certain conditions is on the order of 1 ms (*Berry et al., 1997*; *Reich et al., 1997*; *Berry and Meister, 1998*; *Keat et al., 2001*; *Uzzell and Chichilnisky, 2004*) and that downstream mechanisms could potentially read out these temporally precise RGC signals (*Alonso et al., 1996*). It remains unclear from these studies exactly how this high temporal precision would be important for vision. Studies of readout of RGC signals from the macaque retina have shown that for reconstruction of images from responses to flashed stimuli, and for speed and direction discrimination with moving stimuli, ~10 ms temporal resolution of readout from RGC signals is optimal (*Chichilnisky and Kalmar, 2002*; *Frechette et al., 2005*; *Brackbill et al., 2020*; *Wu et al., 2024*). Nonetheless, finer temporal precision could be part of the neural code of RGCs under certain visual stimulus conditions, such as compensation for fixational eye drift (*Wu et al., 2024*). In sum, much evidence supports the idea that the temporal resolution of RGC signal readout in the brain is likely to be on the order of tens of milliseconds for many visual tasks, but this may not be true for all conditions.

Another important assumption is that visual sensations produced in the brain are based on linear reconstruction of the incident image from retinal inputs. This is a first-order approximation to facilitate real-time optimization of electrical stimulation, and is almost certainly wrong in detail. A more accurate model could involve replacing this with a nonlinear and/or biologically realistic reconstruction

(*Parthasarathy et al., 2017*; *Kim et al., 2021*; *Wu et al., 2022*). However, these approaches are far more complex and cannot yet be optimized in real time.

This work also relies on several empirically tested assumptions about electrical stimulation. First, the brief (150 μs) and low (<4 μA) current pulses used in this study typically evoke a single spike with low and precise latencies (e.g. 0.73 ± 0.05 ms *Sekirnjak et al., 2011*), in part because the mechanism of activation is direct depolarization rather than network-mediated excitation. Second, electrical stimuli at the same electrode separated by at least 10 ms generate approximately independent responses (*Talaminos-Barroso et al., 2020*), making temporal dithering possible. Third, distant electrodes generate independent responses (*Figure 5*), making spatial multiplexing possible. While these electrical stimulation properties may be widely applicable, they should be tested and quantified in each neural circuit before applying the temporal dithering and spatial multiplexing approach.

Finally, the approach relies on the assumption that it is possible to deliver a sufficiently large number of electrical stimuli within a visual integration time to produce high quality artificial vision. The total number of electrical stimuli required depends on the number of cells targeted, their expected firing rates for the visual image, and distribution of RGC activation probabilities in the electrical stimulus dictionary. Future work should identify how these factors vary across individuals, species, and neural circuits.

## Extensions and broader applicability of the proposed approach

The modular nature of the dynamic optimization approach enables several potential extensions. The single-electrode dictionary could be enhanced with multi-electrode stimulation patterns designed to optimize cellular selectivity, response diversity, or ideally the overall expected algorithm performance (*Jepson et al., 2014*; *Fan et al., 2019*; *Vilkhu et al., 2021*; *Vasireddy et al., 2023*). Additionally, the efficient and real-time greedy algorithm could be replaced with an algorithm that identifies the optimal stimulation sequence for multiple time steps, perhaps accounting for predicted future saccade locations. Finally, each module could be optimized for metrics such as performance, hardware efficiency (e.g. *Figure 6B*), or stability/robustness for chronic function.

The greedy temporal dithering and spatial multiplexing approach relies on the ability to efficiently compute and deliver optimal electrical stimuli in a bidirectional electronic implant. Although this procedure exceeds the capabilities of current devices, two features of the approach support implementation on implantable hardware with a limited size and power budget.

First, the hardware requirements of the proposed closed-loop procedure – calibration of single-electrode stimuli followed by dynamically optimized stimulation for encoding visual scenes – are less stringent than those of a real-time closed-loop system. In the latter, one records and analyzes the results of every electrical stimulus to determine the stimulus for the next time step. Instead, the present approach relies only on identifying the *average* electrical response properties of each cell in advance. After this initial calibration, the stimulation sequence is decided in an open-loop manner by optimizing the *expected* visual reconstruction associated with each electrical stimulus.

Second, the approach can effectively exploit non-selective activation. Selective activation of every cell in a region of the retina, if achievable, would make it possible to create arbitrary patterns of neural activity, but in practice this is difficult with real neural interfaces. The presented framework uses all available stimulation patterns as efficiently as possible by directly optimizing for stimulus reconstruction. In fact, the approach frequently exploits non-selective activation, in order to evoke desired spiking activity in fewer time steps (*Figure 6*).

## Translational potential

The physiological similarities between human and macaque retina (*Cowan et al., 2019*; *Kling et al., 2020*; *Soto et al., 2020*; *Rodieck, 1998*), including their responses to electrical stimulation (*Madugula et al., 2022*), suggest that the benefits of the present stimulation approach could translate from the ex vivo lab prototype with healthy macaque retina to an in vivo implant in the degenerated human retina. However, several technical innovations are required to enable chronic in vivo recording and stimulation. First, new surgical methods must be developed to implant a tiny and high-density chip on the surface of the retina with close and stable contact, to create a lasting, stable interface with RGCs. Second, modifications to the dynamic optimization approach are necessary to mitigate non-stationarities in electrical response properties, which are common in chronic recordings with

multi-electrode array implants (*Perge et al., 2013*; *Downey et al., 2018*). Third, receptive field locations, cell types, and reconstruction filters must be inferred using spike waveform features and spontaneous activity rather than light-evoked responses in a blind retina (*Sekirnjak et al., 2011*; *Li et al., 2015*; *Richard et al., 2015*; *Zaidi et al., 2022*). Fourth, the stimulation approach must be modified to account for changes in spontaneous/oscillatory activity in the degenerated retina (*Sekirnjak et al., 2011*; *Goo et al., 2015*; *Trenholm and Awatramani, 2015*). Finally, the approach must be tested with the visual and electrical properties in the central retina (*Gogliettino et al., 2023*), the most clinically relevant location for a retinal implant.

The present approach could also leverage other methods developed for improving the performance of existing, low-resolution implants. Examples of these methods are context-dependent image preprocessing (*Cha et al., 1992*; *McCarthy et al., 2011*; *Lieby et al., 2011*; *Vergnieux et al., 2017*; *Ho et al., 2019*), limiting to sparse stimulation (*Loudin et al., 2007*), exploiting the adaptation of sensory systems (*Rouger et al., 2007*; *Merabet and Pascual-Leone, 2010*), and exploiting perceived phosphenes due to axon bundle activation for optimizing stimulation (*Granley et al., 2022*; *de Ruyter van Steveninck et al., 2022*; *Relic et al., 2022*). Ideally, a unified framework such as the one presented here would include these and potentially other approaches to optimal stimulation.

Electrical stimulation of the visual cortex has also been tested for vision restoration (*Chen et al., 2020*), and one study (*Beauchamp et al., 2020*) deployed a dynamic stimulation approach that demonstrated impressive performance in human participants. However, both studies only considered simple visual stimuli which can be described by lines (such as English letters and numbers) or a few dots. The dynamic optimization framework presented here could provide a way to precisely control neural activity for arbitrarily complex stimuli and improve the performance of a range of cortical sensory implants.

## Methods
### Retinal preparation
Extracellular multi-electrode recording and stimulation in macaque retina were performed as described previously (*Jepson et al., 2013*; *Grosberg et al., 2017*). Briefly, eyes were obtained from terminally anesthetized macaque monkeys used for experiments in other laboratories, in accordance with Institutional Animal Care and Use Committee guidelines. After enucleation, the eyes were hemisected and the vitreous humor was removed. The hemisected eye cups were stored in oxygenated bicarbonate-buffered Ames' solution (Sigma) during transport to the laboratory. The retina was then isolated from the pigment epithelium under infrared illumination and held RGC side down on a custom multi-electrode array (see below). Throughout the experiments, the retina was superfused with Ames' solution at 35°C.

For experimental validation of temporal dithering, eyes were obtained from adult Long-Evans rats in accordance with Institutional Animal Care and Use Committee guidelines. Immediately after enucleation, the anterior portion of the eye and vitreous humor were removed under infrared illumination and the eye cup placed in oxygenated bicarbonate-buffered Ames' solution. The retina was then isolated under infrared illumination and held RGC side down on a custom multi-electrode array. Throughout the experiments, the retina was superfused with Ames' solution at 31°C.

### Electrical recordings
A custom 512-electrode stimulation and recording system (*Hottowy et al., 2008*; *Hottowy et al., 2012*) was used to deliver electrical stimuli and record spikes from RGCs. The electrodes were organized in a 16 × 32 isosceles triangular lattice arrangement, with 60 μm spacing between electrodes (*Litke et al., 2004*). Electrodes were 10 μm in diameter and electroplated with platinum black. For recording, raw voltage signals from the electrodes were amplified, filtered (43–5000 Hz), and multiplexed with custom circuitry. These voltage signals were sampled with commercial data acquisition hardware (National Instruments) at 20 kHz per channel. For recording and stimulation, a platinum ground wire circling the recording chamber served as a distant ground.

## Electrical stimulation

For electrical stimulation, custom hardware (*Hottowy et al., 2012*) was controlled by commercial multifunction cards (National Instruments). Current was passed through each of the 512 electrodes individually, with 40 different amplitudes (0.1–4 µA, logarithmically spaced), 27 times each. For each amplitude, charge-balanced triphasic current pulses with relative amplitudes of 2:−3:1 and phase widths of 50 µs (total duration 150 µs) were delivered through the stimulating electrode (amplitude corresponds to the magnitude of the second, cathodal phase of the pulse). This pulse shape was chosen to reduce stimulation artifacts in the recordings. Custom circuitry disconnected the recording amplifiers during stimulation, reducing stimulation artifacts and making it possible to identify elicited spikes on the stimulating electrode as well as nearby electrodes (*Hottowy et al., 2012*; *Jepson et al., 2013*).

## Visual stimulation

Recordings obtained with visual stimulation were analyzed to identify spike waveforms of distinct RGCs recorded, using spike sorting methods described previously (*Field and Chichilnisky, 2007*; *Litke et al., 2004*). Specifically, the spike times of each cell were typically identified using relatively large spikes detected near the soma. Then, the complete spatiotemporal signature of the spikes from each cell over all electrodes (the *electrical image*) was computed by averaging the voltage waveforms on all electrodes at and near the times of its recorded spikes (*Litke et al., 2004*). This electrical image provided a template of the cell's spatiotemporal spike waveform, which was then used to identify spikes evoked from cells by electrical stimulation.

Distinct RGC types were identified by their visual responses to a 30-min long white noise stimulus (80 × 40 pixel grid, ~44 µm pixels, refresh rate 120 Hz, low photopic light level). After the stimulus presentation the average stimulus that preceded a spike in each RGC was computed, producing the spike-triggered average (STA) stimulus (*Chichilnisky and Kalmar, 2002*). The STA summarizes the spatial, temporal, and chromatic properties of light responses. Spatial receptive fields were obtained using the spatial sensitivity profile of the STA (*Chichilnisky and Kalmar, 2002*). Features of the STA were used to segregate functionally distinct RGC types (*Rhoades et al., 2019*). For each identified RGC type, the receptive fields formed a regular mosaic covering the region of retina recorded (*Devries and Baylor, 1997*; *Field and Chichilnisky, 2007*), confirming the correspondence to a morphologically distinct RGC type (*Dacey, 1993*; *Wässle et al., 1981*), and in some cases revealing complete recordings from the population. The density and light responses of the five most frequently recorded RGC types uniquely identified them as ON and OFF midget, ON and OFF parasol and small bistratified cells. Subsequent analysis was restricted to two numerically dominant RGC types in the macaque retina – ON and OFF parasol cells – which were sampled efficiently in our experiment and formed nearly complete mosaics covering the region recorded. In the rat retina, analysis was restricted to two RGC types – putative ON brisk transient and OFF brisk transient cells identified by their receptive fields and spiking autocorrelation (*Ravi et al., 2018*) – which formed reasonably complete mosaics covering the region recorded and had properties broadly resembling those of ON and OFF parasol cells in the macaque retina.

## Temporal dithering algorithm

Given the stimulus reconstruction filter and electrical stimulation dictionary, the goal of the greedy temporal dithering algorithm is to identify a sequence of electrical stimuli that encodes a target visual stimulus.

Let $s \in (pixels)$ be the target visual stimulus, $A \in (pixels \times cells)$ be the stimulus reconstruction filter, and $r_i \in (cells)$ be the observed response vector (consisting of a zero or a one for each cell) produced in the population of cells stimulated using electrical stimulation dictionary element $c_i$ with the associated probability vector $p_{c_i} \in (cells)$.

Multiple dictionary elements must be combined to generate rich spatiotemporal population responses that capture the visual information in a target visual stimulus. We therefore define the objective as finding a sequence of dictionary elements $\{c_1, \cdots, c_T\}$ that minimizes the expected mean squared error between the target visual stimulus and the image reconstructed from the sequence of responses $\{r_1, \cdots, r_T\}$. The responses are stochastic with $r_i \sim Bernoulli\ (p_{c_i})$ and we assume that responses are generated independently across time steps. The resulting objective function is:

$$c_1, \ldots, c_T = argmin \, E_{r_i \sim Bernoulli \, (p_{c_i})} \, \| \, s \, - \, A \, (r_1 \, + \ldots + \, r_T) \, \|^2$$

To efficiently solve this optimization problem, the right-hand side can be decomposed into bias and variance terms as follows:

$$E_{r_i \sim Bernoulli \, (p_{c_i})} \, \| \, s \, - \, A \, (p_{c_1} \, + \ldots + \, p_{c_T}) \, + \, A \, (p_{c_1} \, + \ldots + \, p_{c_T}) \, - \, A \, (r_1 \, + \ldots + \, r_T) \, \|^2 \tag{2}$$

$$= \, \| \, s \, - \, A \, (p_{c_1} \, + \ldots + \, p_{c_T}) \, \|^2 \, + \, E \, \| \, A \, (p_{c_1} \, + \ldots + \, p_{c_T}) \, - \, A \, (r_1 \, + \ldots + \, r_T) \, \|^2 \tag{3}$$

$$= \, \| \, s \, - \, A \, (p_{c_1} \, + \ldots + \, p_{c_T}) \, \|^2 \, + \, E \, \| \, A \, (p_{c_1} \, - \, r_1) \, \|^2 \, + \ldots + \, E \, \| \, A \, (p_{c_T} \, - \, r_T) \, \|^2 \tag{4}$$

$$= \, \| \, s \, - \, A \, (p_{c_1} \, + \ldots + \, p_{c_T}) \, \|^2 \, + \, tr \, (Cov \, (Ar_1)) \, + \cdots + \, tr \, (Cov \, (Ar_T)) \tag{5}$$

where *Equation 2* follows from adding and subtracting $A \, (p_{c_1} \, + \ldots + \, p_{c_T})$; *Equation 3* expands the square of summation and uses the fact that $EAr_i \, = \, Ap_{c_i}$ to zero out the cross term; and *Equation 4* uses the fact that neural responses at different time steps are independent of each other.

The expression $(tr \, ( \, Cov \, (Ar_i)))$ corresponds to the total variance across all pixels of the visual image for the dictionary element $c$ chosen at step $i$. When cells respond independently to electrical stimulation, this variance term simplifies to $\sum_n \|a_n\|^2 p_{n,c} \, (1 \, - \, p_{n,c})$, where $p_{n,c}$ is the activation probability of cell $n$ with dictionary element $c$, and $a_n$ is the reconstruction filter for cell $n$ (the $n$th column of $A$).

Below, we present two methods to solve the above optimization problem and two additional methods to solve relaxed optimization problems for comparison purposes. The first method provides a greedy solution which can be deployed in real time and handles dependencies between successive stimuli. The second method provides an upper bound on the optimal solution by jointly optimizing a vector of the number of times each dictionary element is used. A third method relaxes the above objective to provide an upper bound on performance with no electrical stimulation constraints by directly optimizing the number of spikes for each cell. Finally, a fourth method approximates the function of present-day retinal implants by optimizing a mapping between visual stimulus intensity and current amplitude for each electrode.

For the last three methods which generate solutions irrespective of the order of stimulation, a temporal dithering strategy for ordering the electrical stimuli was assumed so that the stimuli would not interfere with one another. The performance for all methods was evaluated in the same manner, by linearly reconstructing the target stimulus from the identified electrical stimulation sequence using samples drawn from the single-electrode calibration data.

## Greedy optimization

Instead of jointly optimizing for the whole stimulation sequence, which is difficult, a greedy approach is used for efficiency. This approach optimizes the choice of stimulation at time step $t$ after fixing the stimulation sequence up to step $t-1$. The choice of dictionary element $c_t$ at time step $t$ is only affected by the first and last terms of *Equation 5*. Hence, the greedy objective function for choosing the dictionary element at time step $t$ is given by:

$$c_t = argmin_{c \in D_t} \| s - A \, (p_{c_1} \, + \ldots + p_{c_{t-1}} + p_c) \, \|^2 + \sum_n \|a_n\|^2 p_{n,c} \, (1 - p_{n,c})$$

Instead of using a fixed dictionary of stimulation patterns $D$ for all time steps, biological and hardware constraints on the stimulation sequence can be incorporated by changing the dictionary elements $D_t$ available at each time step. For example, interactions between time steps produced by refractoriness were avoided by removing dictionary elements that activate cells with probability >0.1 if those cells were targeted with probability >0.1 in the last 100 steps.

## Approximate joint optimization

Instead of selecting the dictionary elements step by step, the optimal number of times each dictionary element should ideally be selected (irrespective of the order of stimulation) can be identified by reformulating the objective function as follows:

$$minimize_w \, \| s - ADw \|^2 + v^T w \, such \, that \, w \, \in Z_+$$

Here, $w \in (stimuli)$ is a vector of non-negative integers corresponding to the number of times each dictionary element is stimulated, $D \in (cells \times stimuli)$ is a matrix of activation probabilities for each cell and dictionary element, and $v \in (stimuli)$ is the variance in decoding corresponding to each dictionary element (given by $v_c = \sum_n \|a_n\|^2 p_{n,c} (1 - p_{n,c})$).

The optimization problem is NP complete due to the integer constraints on $w$. However, it can be approximately solved by relaxing the integer constraint, resulting in the following optimization problem and upper bound on performance:

$$minimize_w \|s - ADw\|^2 + v^T w \text{ such that } w \geq 0$$

## Perfect control optimization

To give an estimate of the best possible reconstruction when there is no constraint on electrical stimulation, the number of spikes for each cell can be directly optimized (irrespective of the order of stimulation) using the following objective function:

$$minimize_{r \geq 0} \|s - Ar\|^2 \text{ such that } r \in Z_+$$

Here, $r \in (cells)$ is a vector of non-negative integers corresponding to the number of times each cell spike. Again, this optimization problem can be approximately solved by relaxing the integer constraint on $r$ resulting in the following optimization problem and upper bound on performance:

$$minimize_r \|s - Ar\|^2 \text{ such that } r \geq 0$$

## Static pixel-wise optimization

To approximate the function of present-day retinal implants, a mapping was learned between the intensity of the visual stimulus near each electrode and the intensity of the current passed through that electrode, to determine which electrical stimuli to deliver (irrespective of the order of stimulation).

First, an affine transformation mapped the visual stimulus onto the electrode array. Second, the average visual stimulus intensity was identified over an approximately 130 μm × 130 μm region around the electrode location. Third, the average visual stimulus intensity on the electrode ($s$) was mapped to the current amplitude ($i$) using a scaled sigmoid:

$$i = a + b/(1 + exp(cs + d))$$

This electrical stimulus was then delivered $n$ times at that electrode. All parameters $\{a, b, c, d, n\}$ for each electrode were simultaneously optimized to minimize reconstruction error across a training set of random checkerboard images.

## Analysis of temporal dithering using calibrated responses

Greedy temporal dithering was analyzed in *Figures 2, 3, and 6* using calibrated responses to visual stimulation (stimulus reconstruction filter) and electrical stimulation (dictionary of single-electrode response probabilities).

The stimulus reconstruction filter for each RGC was approximated using the scaled receptive field (*Brackbill et al., 2020*). Briefly, the receptive field was obtained by computing the spatial component of the rank 1 approximation of the STA. The receptive field was then denoised by computing the robust standard deviation ($\sigma$) of the magnitudes of all pixels, zeroing out pixels with absolute value less than $2.5\sigma$, and retaining the largest spatially contiguous component. Finally, the receptive fields were scaled such that a linear-rectified response model most accurately predicted the average spike count recorded within 50 ms of the onset of static flashed checkerboard stimuli. Note that because the stimulus reconstruction filter is proportional to the receptive field, this approximation matches an optimal linear decoder only if the receptive fields of the cells are orthogonal. This is approximately true: RGC receptive fields form a uniform mosaic sampling the visual field with little overlap (*Devries and Baylor, 1997*; *Chichilnisky and Kalmar, 2002*).

Response probabilities for each single-electrode stimulation pattern were identified after removing electrical artifacts using custom spike sorting software (*Mena et al., 2017*). Briefly, the spike sorting software estimates the electrical stimulation artifacts by modeling the artifact change across

amplitudes with a Gaussian process, subtracts the estimated electrical artifacts from the recording and then matches the residual spikes to cell waveforms obtained from recordings obtained with a visual stimulus. The cell activation probabilities for each of the 40 different amplitudes were replaced by values from a sigmoid fitted to all levels, and collected into a dictionary. Each element in the dictionary consisted of an electrode, a stimulus current level, and the evoked spike probability for all recorded cells (typically, for any given electrode and current level, only a few nearby cells had nonzero spike probability).

Dictionary elements that involved activating distant cells along their axons were removed due to the unknown receptive field locations and thus uncertain contribution to stimulus reconstruction (*Grosberg et al., 2017*). Briefly, the responses to electrical stimulation were mapped to a collection of weighted graphs, and graph partitioning and graph traversal algorithms were applied to identify axon bundle activity. The focus was on two characteristic features of axon bundle signals: bidirectional propagation, and growth of signal amplitude with stimulation current (*Tandon et al., 2021*).

Finally, only dictionary elements that activated at least one cell with probability at least 0.01 were retained, resulting in 1000–5000 dictionary elements. A single dictionary element that does not activate any cell (probability = 0) was added to allow the greedy algorithm to avoid stimulation when no available stimulation pattern would decrease error.

Once visual and electrical responses were calibrated, the greedy temporal dithering sequence was applied to 20 static black and white random checkerboard targets, each for up to 10,000 steps. Responses were randomly sampled using calibrated dictionary probabilities and used to reconstruct the stimulus. Reconstruction error was reported as the squared error between the target and reconstruction normalized by the target squared (*relative mean squared error*).

## Validation of temporal dithering using experimentally evoked responses

Responses evoked by the greedy temporal dithering approach during a closed-loop experiment with the rat retina were analyzed in *Figure 4*. A 15-min long white noise visual stimulus (80 × 40 pixel grid, ~44 µm pixels, refresh rate 30 Hz, low photopic light level) recording was used to identify cell locations, types, and spike waveforms as described above. Subsequent analysis was restricted to two numerically dominant ON and OFF cell types, which each formed nearly complete mosaics across the array. Next, an optimal linear reconstruction filter was computed for these cells (*Gogliettino et al., 2023*). Briefly, a linear-nonlinear cascade model was used to simulate RGC responses to white noise images by half-rectifying the inner product of the STAs and the visual stimuli. The response model was scaled to predict the average spike count recorded within 250 ms of the onset of static flashed checkerboard stimuli presented to the retina during the experiment. The optimal linear reconstruction filter was computed using linear least-squares regression of the model responses against the stimuli for a training set of checkerboard stimuli with varying pixel sizes (352, 220, 176, 110, 88, and 55 µm), using 10,000 training images each (60,000 images total).

A single-electrode stimulation scan (42 amplitudes at each of the 512 electrodes individually, 0.1–4 µA, linearly spaced, 15 times each) was then used to compute an electrical stimulus dictionary. Response probabilities for each electrical stimulus were identified using a custom template matching approach (*Gogliettino et al., 2023*) and axon bundle activation thresholds were determined by automated methods as described above (*Tandon et al., 2021*). Briefly, the custom template matching approach performs unsupervised clustering on the voltage traces for each stimulation pattern, then iteratively compares the difference signals between the clusters to cell waveforms estimated from visual stimulus recordings. Response probabilities for each stimulation pattern were smoothed with a sigmoid fitted across amplitudes and collected in the dictionary. Dictionary elements that elicited axon bundle activity or had at least one cell with a poor sigmoid fit were permanently removed from the dictionary. A single dictionary element that did not activate any cell was included to avoid stimulation when any real stimulation pattern would increase error.

After calibration of visual and electrical response properties, a greedy stimulation sequence was computed for each of a collection of 20 random checkerboard visual targets (10 × 5 pixel grid, ~352 µm pixels) during the closed-loop experiment using a greedy temporal dithering implementation optimized for speed (*Lotlikar et al., 2023*). This implementation evaluates dictionary elements in parallel across several distinct regions involving disjoint groups of cells in order to speed up selection

of the electrical stimulation sequence. Each stimulation sequence was assembled until the objective could not be decreased further, resulting in 171–288 stimulations. The stimulation sequence for each target was delivered 25 times at an expanded 3-ms stimulation interval to separate the recorded voltage signals from prior and future electrical stimulation artifacts. The expanded stimulation interval was necessary to determine evoked RGC response probabilities for each target, which were identified using the same custom template matching approach used for the single-electrode stimulation scan.

## Characterizing spatial exclusion radius for spatial multiplexing

The spatial exclusion radius was estimated using a bi-electrode stimulation experiment. The initial response dictionary was characterized using single-electrode stimulation as described above. A target cell was chosen, and the activation curve over the standard current range (42 amplitudes, 0.1–4 µA, linearly spaced) was determined for this cell using the electrode that recorded the largest amplitude spike waveform (primary electrode). Stimulation characterization was then repeated on this electrode with equal current passed simultaneously through a secondary electrode, and the changes in the activation curve relative to the original curve were examined. All secondary electrodes within 400 µm of the primary electrode were tested. The single-electrode activation curve was also re-estimated using secondary electrodes more than 800 µm from the primary electrode and not overlapping the axon of the target cell. Evoked spikes were identified using the spike sorting approach described for the validation experiment above.

The two-electrode stimulation produced an activation curve for each electrode pair, from which the activation threshold (estimated current amplitude producing 50% spike probability) was determined. The fractional change from the single-electrode activation threshold was computed for each secondary electrode, revealing the degree to which the presence of a secondary stimulating electrode influences the responses generated by a particular primary electrode. *Figure 5B* summarizes the absolute change in threshold with increasing distance between stimulating electrodes, generated by computing the weighted mean and the resampled standard error of the weighted mean for test pairs near each distance. The weighting for each electrode pair was inversely proportional to the variance of the single-electrode activation threshold for that cell.

## Extension of greedy dithering to natural scenes with eye movements

The greedy temporal dithering approach was extended to natural viewing by modifications to visual stimulus target generation and reconstruction. For a given natural image, a dynamic visual target was generated by simulating eye movements. A sequence of five hundred fixation locations were sampled, preferentially in the high spatial-frequency regions of the image, with a mean duration of 300 ms (SD 100 ms) between saccades. A patch of size 40 × 80 was taken around each saccade location to generate the dynamic visual stimulus. In some cases, fixational eye movements were also simulated by perturbing the fixation location with a brownian motion (3 pixel SD).

The greedy algorithm was modified such that the stimulation choice at each step considered multiple recent frames of the target. The dynamic target was discretized on the display at 120 Hz, and 83 stimulation choices were made within each frame (corresponding to a stimulation every 0.1 ms). To accommodate the dynamic stimulus, the spatial reconstruction filter was replaced with a spatiotemporal reconstruction filter. For efficiency, the spatiotemporal reconstruction filter was modeled as rank 1 (space–time separable), with the identical time course for all cells and opposite polarity for ON and OFF cells. Hence, each evoked spike influences the reconstruction at multiple subsequent time steps. The straightforward extension of the greedy algorithm is then to choose a stimulation pattern at each time step such that it minimizes the total error over multiple time steps.

For a given stimulation sequence, the image is assembled by first reconstructing each frame of the dynamic visual stimulus using the spatiotemporal reconstruction filter. Then, each frame of the reconstructed dynamic stimulus is 'pasted' at the fixation location at the time of the spike. The intensity for each pixel in the final reconstructed image is estimated by averaging the intensity across all fixation locations in which the recorded cells have reconstruction filters that include the pixel.

## Incorporating perceptual similarity metrics

Possible improvements to the approach that could be produced by optimizing perceptual similarity (rather than mean squared error) in the stimulation objective were analyzed after simplifying

modifications. First, instead of image-dependent and random fixation locations, all possible saccade locations were considered. This corresponds to a uniform distribution of fixation locations, and the visual scene is reconstructed by averaging the reconstruction of image patches corresponding to various fixation locations. Next, for each fixation location, the corresponding image patch was reconstructed using *expected* responses (rather than measured, stochastic responses). Note that unlike the algorithm presented above, this formulation does not account for inter-trial variability. Finally, instead of greedily optimizing the stimulation sequence, the number of stimuli for all dictionary elements and fixation locations were jointly optimized. Given these simplifications, the following optimization problem was solved:

$$min_{\{w_i\} \geq 0}\, d\left(s,\, G\left(\{ADw_i\}_{i=1}^{i=\#\,patches}\right)\right) + \lambda \sum_i \|w_i\|_1$$

where $d$ is the measure of similarity, $s \in (pixels)$ is the target visual stimulus, $A \in (pixels \times cells)$ is the reconstruction filter, $D \in (cells \times stimuli)$ is the dictionary, $w_i \in (stimuli)$ is the number of times each dictionary element is stimulated for patch $i$, and $G$ is an operator that averages the reconstruction of individual patches to assemble the entire image. To explore the reconstruction under different stimulation budgets, $\lambda$ is varied to penalize stimulating a large number of dictionary elements.

## Acknowledgements

We thank J Carmena, K Bankiewicz, T Moore, W Newsome, M Taffe, T Albright, E Callaway, H Fox, R Krauzlis, S Moriarty, and the California National Primate Research Center for access to macaque retinas. We thank the Stanford Artificial Retina team for helpful discussions. We thank ALS Association Milton Safenowitz fellowship (NPS), NSF Graduate Research Fellowship Grant No. 2146755 and NSF Grant No. 1828993 (AJP), NIH NEI F30-EY-030776-03 (SM), NIH NIMH T32MH-020016, NIH NEI F31-EY-033636, the Fondation Bertarelli, the Stanford Neurosciences Graduate Program (AG), Polish Academy of Sciences DEC-2013/10/M/NZ4/00268 (PH), Research to Prevent Blindness Stein Innovation Award, Wu Tsai Neurosciences Institute Big Ideas, NIH NEI R01-EY021271, NIH NEI R01-EY029247, NIH NEI P30-EY019005, and NSF/CRCNS grant (EJC) for funding this work.

## Additional information

### Funding

| Funder | Grant reference number | Author |
|---|---|---|
| ALS Association | | Nishal Pradeepbhai Shah |
| National Science Foundation Graduate Research Fellowship | 2146755 | AJ Phillips |
| National Science Foundation | 1828993 | AJ Phillips |
| National Eye Institute | F30-EY-030776-03 | Sasidhar Madugula |
| National Institute of Mental Health | T32MH-020016 | Alex R Gogliettino |
| National Eye Institute | F31-EY-033636 | Alex R Gogliettino |
| The Fondation Bertarelli | | Alex R Gogliettino |
| The Stanford Neurosciences Graduate Program | | Alex R Gogliettino |
| Polish Academy of Sciences | DEC-2013/10/M/NZ4/00268 | Pawel Hottowy |
| National Eye Institute | R01-EY021271 | EJ Chichilnisky |

| Funder | Grant reference number | Author |
|---|---|---|
| National Eye Institute | R01-EY029247 | EJ Chichilnisky |
| National Eye Institute | P30-EY019005 | EJ Chichilnisky |
| National Science Foundation | NSF/CRCNS | EJ Chichilnisky |

The funders had no role in study design, data collection, and interpretation, or the decision to submit the work for publication.

## Author contributions

Nishal Pradeepbhai Shah, Conceptualization, Resources, Data curation, Software, Formal analysis, Supervision, Validation, Investigation, Visualization, Methodology, Writing – original draft, Project administration, Writing – review and editing; AJ Phillips, Conceptualization, Resources, Data curation, Software, Formal analysis, Validation, Investigation, Visualization, Methodology, Writing – original draft, Writing – review and editing; Sasidhar Madugula, Investigation, Methodology, Writing – review and editing; Amrith Lotlikar, Resources, Data curation, Software, Investigation; Alex R Gogliettino, Lauren Grosberg, Jeff Brown, Pulkit Tandon, Pawel Hottowy, Alexander Sher, Alan M Litke, Methodology, Writing – review and editing; Madeline Rose Hays, Resources, Software, Investigation, Methodology; Aditya Dusi, Investigation, Writing – review and editing; Wladyslaw Dabrowski, Resources; Subhasish Mitra, Supervision, Writing – review and editing; EJ Chichilnisky, Conceptualization, Resources, Supervision, Funding acquisition, Validation, Investigation, Methodology, Writing – original draft, Project administration, Writing – review and editing

## Author ORCIDs

Nishal Pradeepbhai Shah  https://orcid.org/0000-0002-1275-0381
AJ Phillips  https://orcid.org/0000-0001-9109-6551
Alan M Litke  https://orcid.org/0000-0003-3973-3642
EJ Chichilnisky  https://orcid.org/0000-0002-5613-0248

## Decision letter and Author response

Decision letter https://doi.org/10.7554/eLife.83424.sa1
Author response https://doi.org/10.7554/eLife.83424.sa2

# Additional files

## Supplementary files

• MDAR checklist

## Data availability

Data and code are available on Dryad at https://doi.org/10.5061/dryad.pk0p2ngrv.

The following dataset was generated:

| Author(s) | Year | Dataset title | Dataset URL | Database and Identifier |
|---|---|---|---|---|
| Shah N, Phillips AJ, Madugula S, Lotlikar A, Gogliettino A, Hays M, Grosberg L, Brown J, Dusi A, Tandon P, Hottowy P, Dabrowski W, Sher A, Litke A, Mitra S, Chichilnisky EJ | 2024 | Data from: Precise control of neural activity using dynamically optimized electrical stimulation | https://doi.org/10.5061/dryad.pk0p2ngrv | Dryad Digital Repository, 10.5061/dryad.pk0p2ngrv |

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
