## [Editor Report]

This valuable study proposes a new algorithm for determining the electrical stimulation delivered through a sensory-neural/retinal implant with the aim of improving the perceptual benefit to implant users. The evidence supporting the conclusions is solid, with additional experiments and analyses submitted during the revision having significantly strengthened the study. The work will be of interest to both neuroscientists and neuroengineers.

---

## [Decision Letter]

**Decision letter after peer review:**

Thank you for submitting your article "Precise control of neural activity using temporally dithered and spatially multiplexed electrical stimulation" for consideration by *eLife*. Your article has been reviewed by 3 peer reviewers, one of whom is a member of our Board of Reviewing Editors, and the evaluation has been overseen by Joshua Gold as the Senior Editor. The following individual involved in the review of your submission has agreed to reveal their identity: Mohit Shivdasani (Reviewer #3).

Essential revisions:

Included here is a brief evaluation summary and list of revisions the reviewers and review editor deem essential for the authors to address. The public summaries and full, individual reviewers' recommendations for the authors are also appended below. The authors are advised to address the public summaries briefly, and the individual recommendations in a detailed, point-by-point manner.

As you will be able to read below, reviewers appreciated the importance of the study and its potentially broad interest. The approach to formulating the problem of choosing electrical stimuli for visual prostheses as a data-driven optimization problem holds promise for several sensory-neural prostheses. The writing was relatively clear, the figures appropriate, and the methods mostly rigorous. However, reviewers raised concerns with regard to some of the claims made, particularly as it pertains to the full greedy, dithering, multiplexed algorithm and its potential to greatly improve the quality of vision delivered by a retinal implant. The key points that need to be addressed can be summarized as follows:

1) Please provide more experimental data (specifically: reconstructed images and reconstruction errors) to substantiate the claim that the algorithm can improve the quality of vision in an ex vivo setting. The main evidence that is presented about the quality of vision that might be achieved is a computer simulation; that is, the image reconstructions and reconstruction errors given in Figures 2 & 3 with the dithered, but not multiplexed version of the algorithm. However, the same cannot be said about the ex vivo experiment. While the outputs of the dithered & multiplex version were indeed applied to ex vivo retina, the output is all simulated and no experimental validation data is presented. However, it is possible that the experimentally observed retinal output might differ from (the assumption of) a linear sum of dictionary elements. All reviewers agreed that if the authors could report on the results of the experiments in which algorithmic stimulation is applied to ex vivo retina, and report this in terms of image reconstructions and reconstruction error, this would greatly improve the strength of evidence.

2) Please expand the discussion on the theoretical assumptions regarding visual processing in the brain and the perception of phosphenes through electrical stimulation that the study is based on, as it may limit the translational impact of the work. All reviewers agreed that the study relies on several significant assumptions about neural coding in the retina, the visual brain, and interactions between electrodes, some of which have been recently challenged. This includes the assumption that neural coding in the retina is solely based on a firing-rate code, that visual perception is solely based on the number of spikes within a slow temporal integration window, and that non-simultaneous interleaved electrical stimulation does not lead to neural interactions. At a minimum, the authors should address these limitations clearly in their Discussion, and comment on the potential implications of the failure of these assumptions on their algorithm performance.

3) Please clarify which figures/results are from simulations and which are from experimental data.

*Reviewer #1:*

Shah et al. propose an algorithm to precisely control RGC activation using electrical stimulation, using temporal dithering, and spatial multiplexing. The main assumption is that the brain has perceptual integration windows, during which a visual percept can be built up by stimulating single (or small groups of) neurons in rapid succession. Which electrodes to stimulate to achieve a desired percept is dictated by a dictionary of stimulation patterns. The authors demonstrate the effectiveness of their method on ex vivo recordings of ON and OFF parasol cells.

The biggest strengths of the study are the theoretical contributions and the experimental recordings used to demonstrate the effectiveness of their algorithm. The thinking follows a number of recent efforts in the field to think about visual prosthetic stimulation as a closed-loop data-driven optimization problem. This may have benefits over other open-loop stimulation techniques.

However, the biggest weakness of the study is a reliance on a number of controversial assumptions about the neural code of vision. The first is the existence of a slow temporal integration window during which the brain cannot distinguish the order of stimuli presented and/or sums up RGC activity to decode the presented stimulus. The paper presents only limited (and dated) evidence for this. Second, the assumption of a Bernoulli distribution is at the very least limiting, as neurons may respond with multiple spikes to a stimulation pattern and some spatial features may be encoded by the relative timing of spikes across neurons. Third, even though the temporal dithering may avoid electrical crosstalk, there may still be neuronal crosstalk on longer timescales, thus challenging the independence assumption. Extending the delay between stimuli in order to avoid neuronal crosstalk may severely limit the utility of the proposed algorithm since it would cap the number of stimuli that could be delivered in one of the assumed temporal integration windows.

Even if the assumptions hold, the presented evidence of the ex vivo recordings would need to provide some additional detail before the practical utility of the proposed algorithm could be judged accordingly. Although the study reports reconstruction errors and example reconstructed images for the simulation experiment, the same cannot be said about the ex vivo experiment. In addition, real-world implementation of the algorithm would presumably include not just stimulus delivery but also in-the-loop stimulus decoding, and it is not clear how quickly that could be done. Lastly, the linear filters would have to be estimated in a degenerated retina, where one could not rely on responses to light stimuli. The paper notes that this could be done by considering the spontaneous activity of cells – I can see how that could allow you to distinguish between ON and OFF cells, for instance, but it is not clear to me how that would allow you to determine the linear filter for each cell. For those reasons, it is somewhat hard to judge the practical utility and potential significance of the proposed algorithm.

Recommendations for the authors:

Methods: The assumption of a Bernoulli distribution seems limiting, as neurons may respond with more than one spike. The decoding overall does not take into account that (at least some) visual information may be encoded in the relative timing of spikes across neurons.

p.7 algorithm section: The major underlying assumption here is that temporally dithered stimuli will be linearly integrated by the retina. Dithering may avoid electrical crosstalk, but neurons may exhibit "crosstalk" on longer timescales due to the relatively slow (as compared to electrical stimulation) temporal dynamics of ion channels. The authors seem to be aware of that as they say "presumably" and mention the idea of a perceptual integration window. But it would have been great to refer to some existing (and more recent) literature on the topic (if any).

p.8 algorithm section: Greedy algorithms are often not guaranteed to find the global optimum. Can the authors show that their proposed algorithm does not get stuck in local optima? How is the final optimization performed at time t, given the dictionary elements?

Ex vivo experiments: It would be helpful to see reconstruction errors and reconstructed images, similar to how they were presented for the simulation study. How long does the decoding/stimulus selection take? Presumably, the dictionary is quite large given the number of neurons. Is there a more efficient way to search the dictionary than O(N)? Or how is that done, and how quickly could it be done in a real-world implementation? I am concerned that this step may severely limit the number of stimuli that could be realistically delivered in a temporal integration window.

In all equations, it would help if the authors used proper formatting to label vectors vs. matrices. For example, is the stimulus reconstruction filter in Eq. 1 a vector and the product is elementwise? What is the size of (and what are the rows/columns in) D on page 8? Etc.

*Reviewer #2:*

This study proposes a new algorithm for determining the electrical stimulation delivered through a sensory neural implant with the aim of improving the perceptual benefit to implant users. The algorithm is evaluated using data from an ex vivo prototype of a retinal prosthesis to computer-simulate the retinal responses expected from applying the algorithm and later by applying stimuli from the full temporally dithered, spatially multiplex algorithm to ex vivo retina.

Presently, stimulation algorithms used clinically are calibrated using limited perceptual data from the user. In contrast, the proposed algorithm uses detailed measurements of retinal responses to electrical stimulation to optimize the stimulation. This is achieved by minimizing the error between a target image and a version of that image reconstructed from the evoked response that is predicted by the algorithm based on the detailed measurements. The use of a data-driven, optimization approach is similar to several other recently proposed neural stimulation algorithms (which are not cited by the study). The distinguishing feature of the algorithm proposed in this study is that it seeks to stimulate in a way that minimizes the interactions between electrodes that can occur when stimulating neural responses. This avoids the need for the algorithm to account for such interactions.

Overall, the main advantage of the proposed approach is that it frames the problem of how to deliver perceptually beneficial electrical stimulation with an implant as a closed-loop/data-driven optimization problem. This has the potential to improve over presently used open-loop strategies. It then provides an algorithm for solving this optimization problem to a good approximation. Applying the algorithm using data recorded from ex vivo retina with a prototype implant is a strength. However, the evaluation of the efficacy of the algorithm is limited. In the first instance, it is limited to computer simulation of the retinal response for the version of the algorithm that uses just temporal dithering. While this analysis supports the conclusion that the proposed algorithm could provide improved visual perception relative to the clinical open-loop strategy, much stronger evidence would be provided by applying the optimal stimuli from the proposed algorithm directly to the ex vivo retinal preparation and measuring the retinal response. This approach to testing the algorithm directly to the ex vivo retina is done for the full version of the algorithm that combines spatial multiplexing with temporal dithering. However, in contrast to simulated results, the study does not report on the reconstructed images that result from applying the algorithm to ex vivo retina, nor on the reconstruction errors. This makes it difficult to evaluate the efficacy of the algorithm.

Section: Introduction.

The motivation for using a temporally dithered, spatially multiplexed algorithm to optimize stimulation stems from the desire to minimize the interactions caused by simultaneous stimulation of the electrodes in evoking a neural response. While this is an important strategy to investigate, the interactions are not typically as "complex" as claimed in the manuscript. Indeed, previous studies in several labs (including the Chichilnisky lab) show that these interactions can typically be described by a linear, weighted sum of the electrode currents followed by a simple static nonlinearity to predict the probability of spiking (a small minority of retinal ganglion cells require more complex nonlinear descriptions) [1 -3]. This model and others have been the basis for alternative data-driven, closed-loop stimulation strategies that optimize the stimulation in a way that seeks to take advantage of the interactions between electrodes to improve the spatial resolution of evoked retinal activity through "current steering".

Section: Greedy temporal dithering to replicate neural code.

Data-driven optimization: The data required for the proposed algorithm is of two types. An exhaustive dictionary of response probabilities to single electrode stimulation across all current amplitudes, and a set of responses used to reconstruct the target image from the predicted response to electrical stimulation. For the latter, reconstruction of the image is achieved by applying linear filters to the predicted response. In the study, these linear filters were derived from cells' receptive fields, obtained by measured responses in the retina to light stimulation. It is noted that this would not be possible in a clinical implant, as the retina is degenerate. However, it is not clear how a set of filters would be obtained in this case. The authors mention that distinct cell types can be identified from spontaneous activity. However, this does not explain how receptive field size and location would be estimated in this situation.

The reconstruction of the image is achieved through linear filtering with a matrix A, with columns, A_j, that are the (scaled) receptive field filters (Eq. 1). However, this is only correct if the receptive field filters of the different cells are orthogonal, i.e. the inner product of each pair of receptive fields is zero. More generally, appropriate linear filtering should be performed by applying the pseudo inverse of the transpose of A. This is because the retinal spike rates are being approximated as the inner product of the receptive field and the image (A_j transposed, matrix-multiplied by the image vector), half-wave rectified. For the receptive fields of ON and OFF parasol cells given in the study, it appears that the receptive fields are approximately orthogonal for the two separate populations due to the non-overlapping tiling of the visual field by each population (e.g. Figure 2). However, it is not clear whether this situation would prevail in the blind retina, as the filters have not been specified in the case.

The greedy optimization algorithm is insufficiently explained in the Methods, including the following points:

• A derivation justifying splitting the objective function into the terms due to the mean and variance is required.

• The terminology for the terms tr(var(A R_i)) is not clearly explained. I assume it is the matrix trace of the covariance matrix of the random variable A R_i.

• The assumption of a Bernoulli random variable for the response, i.e. 1 or 0 spikes, is limiting, given there may be multiple spikes in response to electrical stimulation, especially for activation via the retinal network.

• The expression that was derived for the term tr(var(A R_i)) in the case of Bernoulli random variables should be given.

• It is not explained how the algorithm performs the final optimization at time step t, given elements in a restricted dictionary D_t.

Section: Greedy temporal dithering outperforms open loop methods.

The image reconstruction shown in Figure 2 uses 500, 3000, and 10000 electrical stimuli (shown in A, B and C respectively). However, these are unrealistically large numbers of stimuli: given the temporal perceptual window of 50 ms, mentioned in the Introduction as the time over which retinal responses would be perceptually integrated, and the pulse duration of 0.15 ms used in the study, a maximum of 333 stimuli could be applied during the window. Consequently, the use of 3000 and 10,000 electrical stimuli in the simulations provides unrealistic estimates of the degree to which the image can be reconstructed.

A full comparison of the proposed greedy, closed-loop algorithm to the conventional open-loop algorithm is difficult to evaluate based on the results presented. First, the number of electrical stimuli applied in making the comparison (Figure 3H) is not given. However, it seems likely, given the data in Figure 3G that an unrealistically large 10,000 stimuli were used. If instead a realistic 300-400 stimuli were used there may be little difference between the greedy-closed loop algorithm and the conventional open-loop algorithm.

A second limitation is that, in this subsection, the greedy, closed-loop algorithm appears to have only been tested in simulation. E.g. "For random checkerboard visual stimulus targets, the greedy dithering stimulation sequence was calculated, neural responses were sampled using measured response probabilities evoked by the individual selected stimuli, and then the target image was linearly reconstructed from these responses." Given that all the relevant data required to run the algorithm for the ex vivo retina and implant prototype had been collected during the experiment, it is unclear why the algorithm was not applied to test it by directly measuring responses to the algorithm's stimulation. This would have tested a critical assumption of the greedy-temporal dithering algorithm: that the responses to successive stimuli are statistically independent. Instead, the simulation assumes this to be the case.

A third limitation is that the reconstructed image for the conventional open-loop algorithm does not resemble the phosphene images reported by most retinal implant users. Most implant users report predominantly bright, rather than dark, localized phosphenes [4]. The open-loop reconstruction shown in Figure 3d appears to be largely a gray averaging of light and dark phosphenes, likely due to the linear reconstruction method used.

Some details of the implementation of the open-loop strategy are unclear including:

• How the area that was "near" the electrode was selected when calculating the intensity of the visual stimulus.

• How the temporal sequence of the electrodes was chosen. It seems that the open-loop strategy is also likely, temporally dithered, but without the benefit of data-driven optimization.

Section: Greedy temporal dithering is nearly optimal given the interface constraints.

The comparison of the greedy, closed-loop approximately optimal algorithm to truly optimal algorithms is an important comparison in principle. However, again it is not clear if a realistic number of stimulation pulses were used in performing this comparison (i.e. < 400).

Some details of the implementation of the optimal comparison strategy are unclear including:

• The meaning and purpose of the term V^T w in the objective function.

• Whether w>=0 was required after the integer requirement was relaxed in the optimization.

Section: Spatial multiplexing for fitting multiple stimuli in a visual integration window.

The idea to use spatial multiplexing of stimuli to overcome the limitation in the number of stimuli that can be delivered during a perceptual temporal window is a good idea to investigate. The aim is to choose stimuli on different electrodes that affect neural response independently. However, the initial formulation of what is meant by independence is not correct. This is stated as: "For independence to hold, the following condition must be met: if *p*1 is the activation probability of a given cell with stimulation on electrode 1, and p2 is the activation probability of the same cell with electrode 2, then the activation probability with simultaneous stimulation must be *p*1+*p*2." That this is incorrect can be seen because this formulation could give a probability greater than 1. However, the subsequent description of what is actually implemented appears correct. A general, in-principle way of describing what independence means is that if p1 is the probability of stimulating one cell with electrode 1 and p2 is the probability of stimulating a different cell with electrode 2, then the probability of stimulating both cell 1 and cell 2 using simultaneous stimulation with electrodes 1 and 2 is the product of those probabilities, p1.p2.

In contrast to greedy dithering alone, the use of both greedy dithering and spatial multiplexing was tested in a closed-loop experiment by recording responses to stimuli produced by the algorithm. However, the paper does not report on the image reconstructions, nor the reconstruction errors that were obtained.

Instead, the reported results of the greedy dithering-plus-multiplexing (Figure 4) show only that it is possible to select eight multiplexed electrodes with sufficient separation to ensure minimal interference. This could potentially increase the number of electrodes stimulated with the greedy, closed-loop algorithm by a factor of 8, bringing it to around 2,700 stimuli. This is closer to the 3000 electrode stimulations used in Figure 2b that gave errors that approached the asymptotic limit. However, the results in Figure 4 were obtained using stimulation every 2 ms, not every 0.15 ms (= pulse duration). With this limitation, this reduces the number of electrode stimuli to 200 in a 50 ms perceptual window, which again is not likely to give a good reconstruction error according to the simulations.

Other Results sections.

The sections on hardware constraints, naturalistic viewing conditions, and the use of perceptual similarity measures make useful observations about the potential benefits of the optimization framework for algorithmically determining the electrical stimulation.

Discussion.

The discussion covers many important points well. Regarding the translational potential, I would agree that an important point is "First, new surgical methods must be developed to implant a tiny chip on the surface of the retina with stable contact." But add that it must also be in extremely close contact for retinal ganglion cell spikes to be recorded. Further, a very high-density array (~ 60 μm pitch) and associated electronics for both stimulation and recording must be developed which is suitable in size, form factor, and power consumption for clinical use.

References

[1] Jepson, L. H., Hottowy, P., Mathieson, K., Gunning, D. E., Dąbrowski, W., Litke, A. M., & Chichilnisky, E. J. (2014). Spatially patterned electrical stimulation to enhance resolution of retinal prostheses. Journal of Neuroscience, 34(14), 4871-4881.

[2] Lorach, H., Goetz, G., Smith, R., Lei, X., Mandel, Y., Kamins, T.,.… & Palanker, D. (2015). Photovoltaic restoration of sight with high visual acuity. Nature medicine, 21(5), 476-482.

[3] Maturana, M. I., Apollo, N. V., Hadjinicolaou, A. E., Garrett, D. J., Cloherty, S. L., Kameneva, T.,.… & Meffin, H. (2016). A simple and accurate model to predict responses to multi-electrode stimulation in the retina. PLoS Computational Biology, 12(4), e1004849.

[4] Humayun, M. S., Weiland, J. D., Fujii, G. Y., Greenberg, R., Williamson, R., Little, J., et al. (2003) Visual perception in a blind subject with a chronic microelectronic retinal prosthesis. Vision Research, 43, (2573-2581).

Recommendations for the authors:

Overall, it appears that the approach may offer some important benefits for sensory-neural implant users. However, the reporting of results is not sufficiently complete to draw strong conclusions about the potential benefits. In addition to the Public Review, I have some related suggestions below.

Reconstruction model in the blind retina.

• It would be helpful to provide more detail about how the image reconstruction would work in the blind retinas, beyond what is mentioned regarding the identification of ON and OFF retinal ganglion cell type. How would the size and location of receptive fields be estimated?

• The assumptions underlying the reconstruction model should be described, especially with respect to the orthogonality of the receptive field filters. It would be helpful to describe an approach in the methods that do not rely on this assumption, as I describe in my public comments.

Greed optimization algorithm: There are several aspects of this that could be better explained. These include:

• A derivation justifying splitting the objective function into the terms due to the mean and variance is required.

• The terminology for the terms tr(var(A R_i)) is not clearly explained. I assume it is the matrix trace of the covariance matrix of the random variable A R_i.

• The assumption of a Bernoulli random variable, i.e. 1 or 0 spikes, is limiting, given there may be multiple spikes in response to electrical stimulation, especially for activation via the retinal network.

• The expression derived for the term tr(var(A R_i)) in the case of Bernoulli random variables should be given.

• It is not explained how the algorithm performs the final optimization at time step t, given elements in a restricted dictionary D_t.

• It is not explained how to determine the time for which recently used dictionary elements are excluded from current use.

Section: Greedy temporal dithering outperforms open loop methods

Regarding the number of single-electrode stimuli used in image reconstruction, it would be better to place the numbers used in the context of what is possible in the perceptual time window. It would recommend using the value of 333 instead of 500, as this corresponds to the number of 0.15 pulses that could be fit into a 50 ms window. The value of 3000 roughly corresponds to what might be achieved with spatial multiplexing. The value of 10,000 corresponds to the upper limit that is achievable through this algorithm.

I think it would be beneficial to make it clearer that the results in Figure 3 are simulated. It would also strengthen the study to perform validation in ex vivo retina to apply the greedy temporal dithering stimuli to the retina and reconstruct the image from the responses. If there is a good reason not to do this, this should be explained.

It would improve the study if a reconstruction algorithm that provides an image with a better match to the perception of phosphenes by retinal implant users was used. If this cannot be done, it should be discussed as a limitation of the study.

It would be helpful to clarify some details of the implementation of the open-loop strategy including:

• How the area that was "near" the electrode was selected when calculating the intensity of the visual stimulus.

• How the temporal sequence of the electrodes was chosen.

Section: Greedy temporal dithering is nearly optimal given the interface constraints

A realistic number of stimulation pulses should be used in performing this comparison e.g. < 400 for the pure temporal dithering or < 3000 for the spatially multiplexed, temporal dithering.

It would be helpful to clarify some details of the implementation of the open-loop strategy including:

• The meaning and purpose of the term V^T.w in the objective function.

• Whether w>=0 was required after the integer requirement was relaxed in the optimization.

Section: Spatial multiplexing for fitting multiple stimuli in a visual integration window

As described in my public review, the description of independence is not correct. I have suggested an alternative description that I believe accords with what was actually implemented.

It was surprising that the results of the validation experiments on ex vivo retina with the spatially multiplexed, temporally dithered algorithm were not reported more thoroughly. It is important to provide figures showing the image reconstruction that was achieved and the statistics for the reconstruction error.

*Reviewer #3:*

In this study, Shah and colleagues propose an interesting solution to the non-linear interactions caused by simultaneously stimulating multiple electrodes within a retinal implant. Through high-resolution recordings of ON and OFF parasol retinal ganglion cells, the authors demonstrate that a greedy dithering and spatially multiplexed algorithm, which can also work in the presence of saccadic eye movements, is able to faithfully reconstruct images represented by total numbers of spikes in a given time window across multiple retinal ganglion cells. Essentially, Shah and colleagues propose and demonstrate a method to only stimulate single or groups of 8 electrodes at a time from a pre-established dictionary, but then interleave stimulation of multiple electrodes or groups rapidly across the dictionary to additively build an image. Through their very rigorous and elegant ex vivo recordings in 180 ON and OFF parasol cells across four primate retina preparations, the authors compellingly demonstrate that (i) their greedy algorithm performs better than an open loop algorithm, similar to an optimal algorithm considering the interface constraints, and close but not equal to an ideal control using only a single-electrode dictionary; (ii) that groups of electrodes can be simultaneously activated with a high-resolution neural interface without any retinal interactions provided that they are at least 160 μm apart; (iii) that the algorithm performs just as well even with only 50% of the electrodes on the interface and (iv) that the algorithm can work in the presence of saccadic eye movements and performs better when both saccadic and fixational eye movements are made as opposed to saccadic movements alone.

The experimental recordings and performance of the algorithm in various conditions are the biggest strengths of this study and the authors certainly demonstrate that their algorithm can reproduce spiking numbers across an array of cells that resemble closely spiking numbers evoked by visual stimuli for these conditions. In other words, the authors' primary claim that the neural code for visual images in the retina (in the form of spiking numbers) can be faithfully reproduced with electrical stimulation using such an algorithm, is well supported by evidence.

A major weakness in the study however is the reliance of this algorithm on several significant assumptions about neural coding in the retina, neural coding in the visual brain, and interactions between electrodes even with non-simultaneous stimulation. Some of these assumptions have already been highly challenged in several studies in the visual neuroscience field and in studies involving the perception of phosphenes with interleaved stimulation of single electrodes. Therefore, in light of what is currently known about visual encoding and artificial vision, the study whilst showcasing an elegant computational tool perhaps provides only little hope that such an algorithm will actually work in practice to recreate the perception of images with electrical stimulation but instead does lay a foundation for further work to be done with the assessment of future algorithms. The main assumptions that the authors rely on include:

1) That neural coding in the retina is simply based on a number of spikes evoked by populations of cells ignoring any temporal patterns of responses. A plethora of studies has indicated that relative spike timing between groups of retinal ganglion cells for example can encode complex visual features but the greedy algorithm does not aim to mimic these spike timing features.

2) That perception within the brain is solely based on a number of spikes within a slow temporal integration window (the authors cite a 1995 reference for this). Since 1995 though, this has also been challenged, therefore extending the authors' claims of reproducing spike numbers in the retina to reproducing perception in the brain would be contentious.

3) That neural interactions with non-simultaneous interleaved electrical stimulation are absent. There is in silico, electrophysiological and perceptual evidence with retinal implants that interleaving of electrodes still results in neural interactions and that perception with interleaved stimulation with multiple electrodes does not result in a linear summed perception of phosphenes evoked by single electrodes i.e. dictionary elements. Therefore, the algorithm would only work if such interactions are minimal or absent, for example with larger than 0.1 ms intervals between stimulations or more than 160 μm electrode separation. Note, interactions with interleaving also exist with cochlear implants as the current spread is large.

4) That even if the above 3 assumptions were applied and true, the algorithm can faithfully extrapolate to reconstruct moving images at 24 per second. This seems unlikely as presumably the total time required to linearly reconstruct a single static image would extend to many tens or even hundreds of ms given the number of times each dictionary element needs to be accessed to enable reproduction of similar spiking numbers between visual and electrical stimulation, runs in the thousands.

In spite of major reliance on these assumptions, the authors do demonstrate a very useful tool in the form of the greedy algorithm for situations perhaps other than the visual system, where perception with artificial stimulation may be more predictable and interactions with non-simultaneous stimulation may be simpler.

Recommendations for the authors:

It may be possible to address at least some of the limitations in particular (1) and (4) mentioned in the public review. For limitation (1), the authors could try and experiment with their algorithm and reanalyse data to examine if and how well spike timing features (perhaps relative first spike latencies between RGCs or other temporal patterns of spikes) are reproducible. For limitation (4) the authors could at least perform calculations of time taken by the algorithm in each of the situations and targets presented, to examine if these times are realistic.

For limitations (2) and (3), the authors at a minimum should address these clearly in their discussion and the potential implications of the failure of these assumptions on their algorithm performance.

Other things that the authors should consider is including some example raw data from their retinas before and after artifact subtraction in response to both visual targets and their greedy algorithm as a figure.

---

## [Author Response]

Essential revisions:Included here is a brief evaluation summary and list of revisions the reviewers and review editor deem essential for the authors to address. The public summaries and full, individual reviewers' recommendations for the authors are also appended below. The authors are advised to address the public summaries briefly, and the individual recommendations in a detailed, point-by-point manner.As you will be able to read below, reviewers appreciated the importance of the study and its potentially broad interest. The approach to formulating the problem of choosing electrical stimuli for visual prostheses as a data-driven optimization problem holds promise for several sensory-neural prostheses. The writing was relatively clear, the figures appropriate, and the methods mostly rigorous. However, reviewers raised concerns with regard to some of the claims made, particularly as it pertains to the full greedy, dithering, multiplexed algorithm and its potential to greatly improve the quality of vision delivered by a retinal implant. The key points that need to be addressed can be summarized as follows:1) Please provide more experimental data (specifically: reconstructed images and reconstruction errors) to substantiate the claim that the algorithm can improve the quality of vision in an ex vivo setting. The main evidence that is presented about the quality of vision that might be achieved is a computer simulation; that is, the image reconstructions and reconstruction errors given in Figures 2 & 3 with the dithered, but not multiplexed version of the algorithm. However, the same cannot be said about the ex vivo experiment. While the outputs of the dithered & multiplex version were indeed applied to ex vivo retina, the output is all simulated and no experimental validation data is presented. However, it is possible that the experimentally observed retinal output might differ from (the assumption of) a linear sum of dictionary elements. All reviewers agreed that if the authors could report on the results of the experiments in which algorithmic stimulation is applied to ex vivo retina, and report this in terms of image reconstructions and reconstruction error, this would greatly improve the strength of evidence.

We agree that this was the main missing element in the submitted manuscript and have spent recent months addressing this issue directly with experiments. The new results are given in the new section *Experimental validation of greedy temporal dithering* accompanied by the new Figure 4. In this analysis, we apply greedy dithered sequences to the retina ex vivo and directly compute image reconstructions and errors from the measured, evoked neural responses. We also compare these results to our previous approach of using measured responses from the electrical stimulus calibration phase of the experiment, and show that the experimental results align with these expectations. Due to pandemic-era limitations on the availability of monkey retinas, we performed the new experimental validation in the rat retina, and have thus added sections corresponding to this analysis to *Methods*. We think the addition of these experiments has substantially increased the impact of the paper and are grateful to the reviewers for highlighting its importance. We have also outlined the limitations of this experimental validation in the Discussion.

2) Please expand the discussion on the theoretical assumptions regarding visual processing in the brain and the perception of phosphenes through electrical stimulation that the study is based on, as it may limit the translational impact of the work. All reviewers agreed that the study relies on several significant assumptions about neural coding in the retina, the visual brain, and interactions between electrodes, some of which have been recently challenged. This includes the assumption that neural coding in the retina is solely based on a firing-rate code, that visual perception is solely based on the number of spikes within a slow temporal integration window, and that non-simultaneous interleaved electrical stimulation does not lead to neural interactions. At a minimum, the authors should address these limitations clearly in their Discussion, and comment on the potential implications of the failure of these assumptions on their algorithm performance.

We appreciate the reviewers’ focus on what is assumed/tested in our approach, something we have attempted to be very up-front about. As requested, we have provided additional clarification of our assumptions regarding neural coding in the visual brain and electrical stimulation in the Discussion. Here, we summarize the rationale for the three specific assumptions raised by the reviewers, paralleling the new Discussion text:

Firing rate code in the retina: We agree that in addition to firing rate, other features of the retinal code such as relative latency have been shown to carry information about the visual stimulus in some conditions and species (e.g. (Gollisch & Meister, 2008)). However, firing rate is thought to be the dominant feature of the neural code in the primate visual system (Shadlen & Newsome, 1994), so we made a first-order approximation to focus on it in this work. The importance of the firing rate code is also supported by recent work in which we found that macaque RGC spike counts integrated over ~150 ms provide greater image reconstruction accuracy than spike latencies (Brackbill et al., 2020). Based on this work, we think that while our approach may not capture all of the information normally present in RGC visual signals, it likely captures a large fraction of what is useful for vision. However, we agree that features of RGC responses other than firing rate could be important in certain circumstances (e.g. (Meister, 1996)), and have clarified this in the Discussion.

Visual perception based on slow temporal integration: We appreciate that the reviewers raised this important and subtle point, which has valid arguments on both sides. We try to describe our perspective on it more fully here. There is ample evidence that visual signals in the brain are integrated over tens of milliseconds to produce perception. This evidence ranges from flicker fusion experiments to the time scale of synaptic signal transfer to neurophysiological tests of temporal integration (Borghuis et al., 2019; Samaha & Postle, 2015; Tadin et al., 2010; Wutz et al., 2016). It is also consistent with the widespread use of display technology with ~60-100 Hz refresh rates. This known coarse temporal resolution of vision likely arises from long time constants in phototransduction and synaptic transfer. However, in principle, signals transmitted by RGCs to the brain could have finer temporal precision than signals in the photoreceptors, and visual centers in the brain could be sensitive to the precise timing of RGC spikes (as described in modeling studies, e.g. (Gütig et al., 2013)). Indeed, there is some empirical evidence that the temporal precision of spikes in RGCs can, in certain conditions, be on the order of 1 ms (Berry et al., 1997; Berry & Meister, 1998; Keat et al., 2001; Reich et al., 1997; Uzzell & Chichilnisky, 2004). Furthermore, the idea that downstream mechanisms in the brain could “read out” RGC signals with millisecond temporal precision is supported to a limited degree by empirical studies of precisely correlated activity (e.g. (Alonso et al., 1996)). However, it is unclear from these studies exactly how this high temporal precision would be useful for vision. We have performed several in-depth studies of the temporal resolution of readout of RGC signals from the macaque retina. This work has shown that for reconstruction of images from flashed stimuli, and for speed/direction discrimination with moving stimuli, the optimal temporal resolution of primate RGC signal readout is ~10 ms or coarser (Brackbill et al., 2020; Chichilnisky & Kalmar, 2003; Frechette et al., 2005; Wu et al., 2023). On the other hand, high temporal precision could be part of the neural code of RGCs in other specific visual stimulus conditions. For example, we recently found that image reconstruction from macaque RGC spikes in the presence of fixational eye drift is sensitive to spike timing in the 2-5 ms range (Wu et al., 2023). In sum, much evidence leans toward the idea that the temporal resolution of RGC signal readout in the brain is likely to be on the order of tens of milliseconds for many visual tasks. However, this may not be true for all conditions. We have clarified this in the Discussion.

Neural interactions for non-simultaneous interleaved electrical stimulation: While such interactions have been observed in previous studies of electrical stimulation (Ho et al., 2020; Sekhar et al., 2020; Yoon et al., 2020), the current work relies on very low current levels (<4 µA) delivered in brief pulses (150 µsec) that typically produce a single spike, directly evoked by membrane depolarization, with a latency of a few milliseconds (Sekirnjak et al., 2006, 2008) and no network-mediated activation. This suggests that the primary temporal interactions are likely to be the relative refractory period of neurons (~10 ms). This interpretation is supported by the findings in our new experimental validation. We have clarified these considerations in the Discussion.

3) Please clarify which figures/results are from simulations and which are from experimental data.

None of the figures in the paper are based on simulations of retinal responses – all figures use real recorded data. Importantly, however, Figures 2, 3, 6, 7, and 8 use calibrated responses to single-electrode stimulation to compute the reconstruction that is possible if dithered and multiplexed stimulation is used — the data are from real recordings, but the independence and stability of evoked responses is assumed, and samples are drawn from the measured spike probabilities to compute the reconstruction quality. To test whether these assumptions are valid, the new manuscript also includes closed-loop experiments (Figures 4 and 5) which validate the entire calibration-stimulation pipeline. We have added clarifying text throughout the Results and also summarized this information in the Discussion.

Reviewer #1:1.1) Shah et al. propose an algorithm to precisely control RGC activation using electrical stimulation, using temporal dithering, and spatial multiplexing. The main assumption is that the brain has perceptual integration windows, during which a visual percept can be built up by stimulating single (or small groups of) neurons in rapid succession. Which electrodes to stimulate to achieve a desired percept is dictated by a dictionary of stimulation patterns. The authors demonstrate the effectiveness of their method on ex vivo recordings of ON and OFF parasol cells.The biggest strengths of the study are the theoretical contributions and the experimental recordings used to demonstrate the effectiveness of their algorithm. The thinking follows a number of recent efforts in the field to think about visual prosthetic stimulation as a closed-loop data-driven optimization problem. This may have benefits over other open-loop stimulation techniques.

Thank you for recognizing the strengths of the study.

1.2) However, the biggest weakness of the study is a reliance on a number of controversial assumptions about the neural code of vision. The first is the existence of a slow temporal integration window during which the brain cannot distinguish the order of stimuli presented and/or sums up RGC activity to decode the presented stimulus. The paper presents only limited (and dated) evidence for this.

We agree that this was limited in the original submission, and have clarified this important issue in our response to Essential Revisions 2 paragraph 3 above as well as in changes to the Discussion

1.3) Second, the assumption of a Bernoulli distribution is at the very least limiting, as neurons may respond with multiple spikes to a stimulation pattern and some spatial features may be encoded by the relative timing of spikes across neurons.

Please see our response to Recommendations for Authors 1.8 below.

1.4) Third, even though the temporal dithering may avoid electrical crosstalk, there may still be neuronal crosstalk on longer timescales, thus challenging the independence assumption. Extending the delay between stimuli in order to avoid neuronal crosstalk may severely limit the utility of the proposed algorithm since it would cap the number of stimuli that could be delivered in one of the assumed temporal integration windows.

Please see our response to Recommendations for Authors 1.9 below.

1.5) Even if the assumptions hold, the presented evidence of the ex vivo recordings would need to provide some additional detail before the practical utility of the proposed algorithm could be judged accordingly. Although the study reports reconstruction errors and example reconstructed images for the simulation experiment, the same cannot be said about the ex vivo experiment.

We have now performed the key closed-loop validation experiment and provided the results in the paper. Please see our response to Essential Revisions 1 above and the changes to the manuscript.

1.6) In addition, real-world implementation of the algorithm would presumably include not just stimulus delivery but also in-the-loop stimulus decoding, and it is not clear how quickly that could be done.

While the real-time, in-the-loop stimulus decoding (which would use the actual evoked spikes rather than the probability of an evoked spike) could improve performance, it is computationally prohibitive as the reviewer points out. The approach in this paper does not require real-time, in-the-loop stimulus decoding, but instead uses the *expected* stimulus decoding based on a fixed set of calibration measurements. We show that inter-trial variability in total number of spikes is minimal, so that expected spike decoding is sufficient. We have now clarified this in the Discussion.

1.7) Lastly, the linear filters would have to be estimated in a degenerated retina, where one could not rely on responses to light stimuli. The paper notes that this could be done by considering the spontaneous activity of cells – I can see how that could allow you to distinguish between ON and OFF cells, for instance, but it is not clear to me how that would allow you to determine the linear filter for each cell. For those reasons, it is somewhat hard to judge the practical utility and potential significance of the proposed algorithm.

These are excellent points. A recent paper from our group (Zaidi et al., 2023) addresses these issues directly. In that paper, the firing rate and autocorrelation function are first used to classify ON and OFF parasol cell types, as the reviewer suggests. Then, the average linear spatiotemporal filter for each cell type is translated in space to an estimated receptive field location for each recorded cell using its electrical image, which we have shown provides an accurate estimate of its physical location (Li et al., 2015). This procedure was evaluated quantitatively in the aforementioned publication and shown to work well, accurately reproducing the actual measured linear filters of the cells using the autocorrelation and electrical image location. Future work will be needed for additional cell types, including additional electrical features that we measure routinely which can be used to identify more cell types, but the overall approach is expected to be similar. We have clarified this important issue in the Discussion.

Recommendations for the authors:1.8) Methods: The assumption of a Bernoulli distribution seems limiting, as neurons may respond with more than one spike. The decoding overall does not take into account that (at least some) visual information may be encoded in the relative timing of spikes across neurons.

With the low amplitude electrical stimulation that we use (150µs long pulses with peak current amplitude 4µA), we typically observe zero or one directly-evoked spikes for each stimulation pulse (see (Sekirnjak et al., 2006)). This is in part because the pulses are short and the mechanism of activation is direct depolarization (Sekirnjak et al., 2006), rather than a network-mediated excitation. Under these conditions, the Bernoulli assumption is reasonable. We agree that our approach would not translate to other electrical stimulation patterns, such as pulse trains or high current levels or network-mediated activation, which could elicit many spikes. We note that, unlike the present approach, those stimulation paradigms make it difficult/impossible to replicate the neural code, and that evoking one spike at a time is therefore a singular advantage of our approach. We have clarified this in the Discussion.

Please see our response to Essential Revisions 2 paragraph 2 for a discussion of relative spike timing. We also note that with the very high temporal precision of our stimulation (evoked spike time variation of roughly 0.1 ms), if there were some degree of stimulus coding in the relative timing of spikes, that relative timing could certainly be reproduced by the stimulation sequences that we provide, with a suitable modification of the optimization approach.

1.9) p.7 algorithm section: The major underlying assumption here is that temporally dithered stimuli will be linearly integrated by the retina. Dithering may avoid electrical crosstalk, but neurons may exhibit "crosstalk" on longer timescales due to the relatively slow (as compared to electrical stimulation) temporal dynamics of ion channels. The authors seem to be aware of that as they say "presumably" and mention the idea of a perceptual integration window. But it would have been great to refer to some existing (and more recent) literature on the topic (if any).

To clarify, the assumption is *not* that temporally dithered stimuli are linearly integrated by the retina. The assumption is that temporally dithered evoked spikes are linearly integrated by the visual system downstream of the retina, on time scales of tens of ms. The evidence for this is discussed in our response to Essential Revision 2 paragraph 3. In short, while this assumption is not necessarily correct in all conditions, and testing it thoroughly will require an implanted device that does not yet exist, there is ample evidence to support this assumption in many stimulus conditions.

We agree that neuronal “crosstalk” on longer timescales than the temporal dithering is a possibility. Please see our response to Essential Revisions (2) paragraph 4. To avoid the primary temporal interactions due to the relative refractory period of neurons, the stimuli at each timestep are chosen from a ‘valid’ subset of the dictionary that disallows stimulation of any recently targeted cell within its relative refractory period. Our new validation experiments directly test the possibility of crosstalk in the retina ex vivo, and the results are encouraging (Figures 4 and 5). We have now highlighted the importance of independent responses for the temporal dithering approach in the Assumptions subsection of the Discussion.

1.10) p.8 algorithm section: Greedy algorithms are often not guaranteed to find the global optimum. Can the authors show that their proposed algorithm does not get stuck in local optima? How is the final optimization performed at time t, given the dictionary elements?

While we do not show that the greedy algorithm does not get stuck in local optima analytically, we do show empirically that the gap between lower bound on the optimal solution and greedy solution is small (Figure 3, histograms colored orange and blue). This finding indicates that even if the greedy algorithm does sometimes get stuck in local optima, the degradation in performance is insignificant compared to the benefits in reconstruction performance that the algorithm provides.

The formula for the optimization performed at time t over the available set of dictionary elements is given as Equation 5 in the Methods.

1.11) Ex vivo experiments: It would be helpful to see reconstruction errors and reconstructed images, similar to how they were presented for the simulation study.

We agree that direct closed-loop validation with the temporally dithered stimulation is important and have now performed these experiments. Please see our response to Essential Revision (1). In brief, the temporally dithered stimulation conveys very substantial image structure, as predicted using the measurements at the start of the experiment.

1.12) How long does the decoding/stimulus selection take? Presumably, the dictionary is quite large given the number of neurons. Is there a more efficient way to search the dictionary than O(N)? Or how is that done, and how quickly could it be done in a real-world implementation? I am concerned that this step may severely limit the number of stimuli that could be realistically delivered in a temporal integration window.

These are important considerations. The dictionary can indeed potentially be large (the number of elements is equal to the number of stimulation patterns tested), and searching this dictionary could thus be a computationally prohibitive step. We have recently addressed this problem in our group (Lotlikar et al., 2023) using the insight that the greedy search can be decomposed into multiple smaller searches, because far-away electrodes stimulate a disjoint collection of cells. This approach gives a drastic increase in the speed of the algorithm. Other engineering insights (such as finding the right embedded processor, building custom chips, etc.) can further increase the speed. We have decided to not focus on engineering implementation for this conceptual/theoretical paper because it is already fairly long.

1.13) In all equations, it would help if the authors used proper formatting to label vectors vs. matrices. For example, is the stimulus reconstruction filter in Eq. 1 a vector and the product is elementwise? What is the size of (and what are the rows/columns in) D on page 8? Etc.

Thank you for the suggestion. We have clarified vector and matrix dimensions and fixed the equation formatting in the Results.

Reviewer #2 :2.1)This study proposes a new algorithm for determining the electrical stimulation delivered through a sensory neural implant with the aim of improving the perceptual benefit to implant users. The algorithm is evaluated using data from an ex vivo prototype of a retinal prosthesis to computer-simulate the retinal responses expected from applying the algorithm and later by applying stimuli from the full temporally dithered, spatially multiplex algorithm to ex vivo retina.Presently, stimulation algorithms used clinically are calibrated using limited perceptual data from the user. In contrast, the proposed algorithm uses detailed measurements of retinal responses to electrical stimulation to optimize the stimulation. This is achieved by minimizing the error between a target image and a version of that image reconstructed from the evoked response that is predicted by the algorithm based on the detailed measurements. The use of a data-driven, optimization approach is similar to several other recently proposed neural stimulation algorithms (which are not cited by the study). The distinguishing feature of the algorithm proposed in this study is that it seeks to stimulate in a way that minimizes the interactions between electrodes that can occur when stimulating neural responses. This avoids the need for the algorithm to account for such interactions.Overall, the main advantage of the proposed approach is that it frames the problem of how to deliver perceptually beneficial electrical stimulation with an implant as a closed-loop/data-driven optimization problem. This has the potential to improve over presently used open-loop strategies. It then provides an algorithm for solving this optimization problem to a good approximation. Applying the algorithm using data recorded from ex vivo retina with a prototype implant is a strength.

Thank you for summarizing the work and recognizing its strengths. We have referenced additional data-driven optimization approaches for neural stimulation in the manuscript (Choi et al., 2016; Haji Ghaffari et al., 2021; Tafazoli et al., 2020; Vasireddy et al., 2023).

2.2) However, the evaluation of the efficacy of the algorithm is limited. In the first instance, it is limited to computer simulation of the retinal response for the version of the algorithm that uses just temporal dithering. While this analysis supports the conclusion that the proposed algorithm could provide improved visual perception relative to the clinical open-loop strategy, much stronger evidence would be provided by applying the optimal stimuli from the proposed algorithm directly to the ex vivo retinal preparation and measuring the retinal response. This approach to testing the algorithm directly to the ex vivo retina is done for the full version of the algorithm that combines spatial multiplexing with temporal dithering. However, in contrast to simulated results, the study does not report on the reconstructed images that result from applying the algorithm to ex vivo retina, nor on the reconstruction errors. This makes it difficult to evaluate the efficacy of the algorithm.

This is a crucial point. We note that the original analysis was not based on computer simulations, but samples drawn from calibration measurements at the start of each experiment. We have now clarified this, and more importantly added reconstructed images and errors using an actual closed-loop validation experiment as suggested. Please see our response to Essential Revisions 1 above and the changes to the manuscript.

2.3) Section: Introduction.The motivation for using a temporally dithered, spatially multiplexed algorithm to optimize stimulation stems from the desire to minimize the interactions caused by simultaneous stimulation of the electrodes in evoking a neural response. While this is an important strategy to investigate, the interactions are not typically as "complex" as claimed in the manuscript. Indeed, previous studies in several labs (including the Chichilnisky lab) show that these interactions can typically be described by a linear, weighted sum of the electrode currents followed by a simple static nonlinearity to predict the probability of spiking (a small minority of retinal ganglion cells require more complex nonlinear descriptions) [1 -3]. This model and others have been the basis for alternative data-driven, closed-loop stimulation strategies that optimize the stimulation in a way that seeks to take advantage of the interactions between electrodes to improve the spatial resolution of evoked retinal activity through "current steering".

We agree that there has been some progress in understanding interactions during electrical stimulation (though, we emphasize that this is distinct from interactions obtained with *visual* stimulation, where the LN models the reviewer describes have been fairly successful), and indeed some of this work has come from our lab. However, current studies of electrical stimulation, including our own, typically only examine one or a few electrodes at a time, and even in these situations we’ve shown that nonlinear interactions are often more complex than LN (Jepson et al., 2014). Our recent work (not shown) indicates that it occurs more with particular electrode configurations. In related work, we have recently made progress toward a closed-loop calibration strategy that uses “current steering” on 3 neighboring electrodes to improve the selectivity of stimulation (Vasireddy et al., 2023), and this work certainly requires dealing with the substantial nonlinearity that is present in some cases. However, using the above approaches to replicate the complex spatio-temporal pattern of RGC activity with hundreds or thousands of electrodes is far more complex. The approach presented here scales naturally to large numbers of electrodes, and also can include well-calibrated simultaneous-stimulation patterns (e.g. 3-electrode current steering) as “dictionary elements”, within the exact same optimization framework. We have attempted to clarify this a bit more in the Extensions subsection of the Discussion.

2.4) Section: Greedy temporal dithering to replicate neural code.Data-driven optimization: The data required for the proposed algorithm is of two types. An exhaustive dictionary of response probabilities to single electrode stimulation across all current amplitudes, and a set of responses used to reconstruct the target image from the predicted response to electrical stimulation. For the latter, reconstruction of the image is achieved by applying linear filters to the predicted response. In the study, these linear filters were derived from cells' receptive fields, obtained by measured responses in the retina to light stimulation. It is noted that this would not be possible in a clinical implant, as the retina is degenerate. However, it is not clear how a set of filters would be obtained in this case. The authors mention that distinct cell types can be identified from spontaneous activity. However, this does not explain how receptive field size and location would be estimated in this situation.

These points are exactly correct. Please see our response to Public Review 1.7 above.

2.5) The reconstruction of the image is achieved through linear filtering with a matrix A, with columns, A_j, that are the (scaled) receptive field filters (Eq. 1). However, this is only correct if the receptive field filters of the different cells are orthogonal, i.e. the inner product of each pair of receptive fields is zero. More generally, appropriate linear filtering should be performed by applying the pseudo inverse of the transpose of A. This is because the retinal spike rates are being approximated as the inner product of the receptive field and the image (A_j transposed, matrix-multiplied by the image vector), half-wave rectified. For the receptive fields of ON and OFF parasol cells given in the study, it appears that the receptive fields are approximately orthogonal for the two separate populations due to the non-overlapping tiling of the visual field by each population (e.g. Figure 2). However, it is not clear whether this situation would prevail in the blind retina, as the filters have not been specified in the case.

Again, excellent point. Please see our response to Recommendations for Authors 2.22 below.

2.6) The greedy optimization algorithm is insufficiently explained in the Methods, including the following points:A derivation justifying splitting the objective function into the terms due to the mean and variance is required.

We have added this derivation to the Methods.

2.7) The terminology for the terms tr(var(A R_i)) is not clearly explained. I assume it is the matrix trace of the covariance matrix of the random variable A R_i.

Yes, it is the trace of the covariance matrix. We have corrected the term in the Methods.

2.8) The assumption of a Bernoulli random variable for the response, i.e. 1 or 0 spikes, is limiting, given there may be multiple spikes in response to electrical stimulation, especially for activation via the retinal network.

This is a good point that needed to be clarified in the text. Please see our response to Recommendations for Authors 1.8 above.

2.9) The expression that was derived for the term tr(var(A R_i)) in the case of Bernoulli random variables should be given.

We have added this expression in the Methods.

2.10) It is not explained how the algorithm performs the final optimization at time step t, given elements in a restricted dictionary D_t.

We have clarified this point in the Methods.

2.11) Section: Greedy temporal dithering outperforms open loop methods.The image reconstruction shown in Figure 2 uses 500, 3000, and 10000 electrical stimuli (shown in A, B and C respectively). However, these are unrealistically large numbers of stimuli: given the temporal perceptual window of 50 ms, mentioned in the Introduction as the time over which retinal responses would be perceptually integrated, and the pulse duration of 0.15 ms used in the study, a maximum of 333 stimuli could be applied during the window. Consequently, the use of 3000 and 10,000 electrical stimuli in the simulations provides unrealistic estimates of the degree to which the image can be reconstructed.

Please see our response to Recommendations for Authors 2.29 below.

2.12) A full comparison of the proposed greedy, closed-loop algorithm to the conventional open-loop algorithm is difficult to evaluate based on the results presented. First, the number of electrical stimuli applied in making the comparison (Figure 3H) is not given. However, it seems likely, given the data in Figure 3G that an unrealistically large 10,000 stimuli were used. If instead a realistic 300-400 stimuli were used there may be little difference between the greedy-closed loop algorithm and the conventional open-loop algorithm.

We have clarified the number of electrical stimuli applied for the reconstructions in Figure 3. Please see our response to Recommendations for Authors 2.29 below.

2.13) A second limitation is that, in this subsection, the greedy, closed-loop algorithm appears to have only been tested in simulation. E.g. "For random checkerboard visual stimulus targets, the greedy dithering stimulation sequence was calculated, neural responses were sampled using measured response probabilities evoked by the individual selected stimuli, and then the target image was linearly reconstructed from these responses." Given that all the relevant data required to run the algorithm for the ex vivo retina and implant prototype had been collected during the experiment, it is unclear why the algorithm was not applied to test it by directly measuring responses to the algorithm's stimulation. This would have tested a critical assumption of the greedy-temporal dithering algorithm: that the responses to successive stimuli are statistically independent. Instead, the simulation assumes this to be the case.

This is a crucial point. We have now performed the closed-loop validation experiment and shown reconstructions. Please see our responses to Essential Revisions 1 above and the changes to the manuscript.

2.14) A third limitation is that the reconstructed image for the conventional open-loop algorithm does not resemble the phosphene images reported by most retinal implant users. Most implant users report predominantly bright, rather than dark, localized phosphenes [4]. The open-loop reconstruction shown in Figure 3d appears to be largely a gray averaging of light and dark phosphenes, likely due to the linear reconstruction method used.

Another good point. Please see our response to Recommendations for Authors 2.31 below.

2.15) Some details of the implementation of the open-loop strategy are unclear including:How the area that was "near" the electrode was selected when calculating the intensity of the visual stimulus.How the temporal sequence of the electrodes was chosen. It seems that the open-loop strategy is also likely, temporally dithered, but without the benefit of data-driven optimization.

We appreciate the suggestion to clarify. Please see our response to Recommendations for Authors 2.32 below.

2.16) Section: Greedy temporal dithering is nearly optimal given the interface constraints.The comparison of the greedy, closed-loop approximately optimal algorithm to truly optimal algorithms is an important comparison in principle. However, again it is not clear if a realistic number of stimulation pulses were used in performing this comparison (i.e. < 400). Some details of the implementation of the optimal comparison strategy are unclear including:The meaning and purpose of the term V^T w in the objective function.Whether w>=0 was required after the integer requirement was relaxed in the optimization.

We have clarified the number of electrical stimuli applied for the reconstructions in Figure 3. Please see our response to Recommendations for Authors 2.34 below.

2.17) Section: Spatial multiplexing for fitting multiple stimuli in a visual integration window.The idea to use spatial multiplexing of stimuli to overcome the limitation in the number of stimuli that can be delivered during a perceptual temporal window is a good idea to investigate. The aim is to choose stimuli on different electrodes that affect neural response independently. However, the initial formulation of what is meant by independence is not correct. This is stated as: "For independence to hold, the following condition must be met: if p1 is the activation probability of a given cell with stimulation on electrode 1, and p2 is the activation probability of the same cell with electrode 2, then the activation probability with simultaneous stimulation must be p1+p2." That this is incorrect can be seen because this formulation could give a probability greater than 1. However, the subsequent description of what is actually implemented appears correct. A general, in-principle way of describing what independence means is that if p1 is the probability of stimulating one cell with electrode 1 and p2 is the probability of stimulating a different cell with electrode 2, then the probability of stimulating both cell 1 and cell 2 using simultaneous stimulation with electrodes 1 and 2 is the product of those probabilities, p1.p2.

Please see our response to Recommendations for Authors 2.35 below.

2.18) Instead, the reported results of the greedy dithering-plus-multiplexing (Figure 4) show only that it is possible to select eight multiplexed electrodes with sufficient separation to ensure minimal interference. This could potentially increase the number of electrodes stimulated with the greedy, closed-loop algorithm by a factor of 8, bringing it to around 2,700 stimuli. This is closer to the 3000 electrode stimulations used in Figure 2b that gave errors that approached the asymptotic limit. However, the results in Figure 4 were obtained using stimulation every 2 ms, not every 0.15 ms (= pulse duration). With this limitation, this reduces the number of electrode stimuli to 200 in a 50 ms perceptual window, which again is not likely to give a good reconstruction error according to the simulations.

Please see our response to Recommendations for Authors 2.29 below.

2.19) Other results sections.The sections on hardware constraints, naturalistic viewing conditions, and the use of perceptual similarity measures make useful observations about the potential benefits of the optimization framework for algorithmically determining the electrical stimulation.

Thank you.

2.20) Discussion.The discussion covers many important points well. Regarding the translational potential, I would agree that an important point is "First, new surgical methods must be developed to implant a tiny chip on the surface of the retina with stable contact." But add that it must also be in extremely close contact for retinal ganglion cell spikes to be recorded. Further, a very high-density array (~ 60 μm pitch) and associated electronics for both stimulation and recording must be developed which is suitable in size, form factor, and power consumption for clinical use.

For the first point, we have added this to the Discussion as suggested. For the second point, please see the updated section on hardware design in the Discussion.

ReferencesJepson, L. H., Hottowy, P., Mathieson, K., Gunning, D. E., Dąbrowski, W., Litke, A. M., & Chichilnisky, E. J. (2014). Spatially patterned electrical stimulation to enhance resolution of retinal prostheses. Journal of Neuroscience, 34(14), 4871-4881.Lorach, H., Goetz, G., Smith, R., Lei, X., Mandel, Y., Kamins, T.,.… & Palanker, D. (2015). Photovoltaic restoration of sight with high visual acuity. Nature medicine, 21(5), 476-482.Maturana, M. I., Apollo, N. V., Hadjinicolaou, A. E., Garrett, D. J., Cloherty, S. L., Kameneva, T.,.… & Meffin, H. (2016). A simple and accurate model to predict responses to multi-electrode stimulation in the retina. PLoS Computational Biology, 12(4), e1004849.Humayun, M. S., Weiland, J. D., Fujii, G. Y., Greenberg, R., Williamson, R., Little, J., et al. (2003) Visual perception in a blind subject with a chronic microelectronic retinal prosthesis. Vision Research, 43, 2573-2581).

Recommendations for the authors:2.21) Overall, it appears that the approach may offer some important benefits for sensory-neural implant users. However, the reporting of results is not sufficiently complete to draw strong conclusions about the potential benefits. In addition to the Public Review, I have some related suggestions below.Reconstruction model in the blind retina.It would be helpful to provide more detail about how the image reconstruction would work in the blind retinas, beyond what is mentioned regarding the identification of ON and OFF retinal ganglion cell type. How would the size and location of receptive fields be estimated?

Please see our response to Public Review 1.7 above.

2.22) The assumptions underlying the reconstruction model should be described, especially with respect to the orthogonality of the receptive field filters. It would be helpful to describe an approach in the methods that do not rely on this assumption, as I describe in my public comments.

We do indeed approximate the optimal linear reconstruction filters using the measured receptive fields of the cells. We agree that the decoder used is not the ‘inverse’ of the receptive fields, except in the case that the receptive fields of the cells are orthogonal. We can use the standard least-squares solution to move away from the assumption of orthogonality. We have clarified the important point that this is an approximation in the Results and Methods.

2.23) Greedy optimization algorithm: There are several aspects of this that could be better explained. These include:A derivation justifying splitting the objective function into the terms due to the mean and variance is required.

Thank you. We have added this derivation to the Methods.

2.24) The terminology for the terms tr(var(A R_i)) is not clearly explained. I assume it is the matrix trace of the covariance matrix of the random variable A R_i.

Yes, it is the trace of the covariance matrix. We have corrected the term in the Methods.

2.25) The assumption of a Bernoulli random variable, i.e. 1 or 0 spikes, is limiting, given there may be multiple spikes in response to electrical stimulation, especially for activation via the retinal network.

Please see our response to Recommendations for Reviewers 1.8 above.

2.26) The expression derived for the term tr(var(A R_i)) in the case of Bernoulli random variables should be given.

We have added this expression in the Methods.

2.27) It is not explained how the algorithm performs the final optimization at time step t, given elements in a restricted dictionary D_t.

We have clarified this point in the Methods.

2.28) It is not explained how to determine the time for which recently used dictionary elements are excluded from current use.

Thank you for pointing out this missing information. We exclude dictionary elements to disallow stimulation of any recently targeted cell for 100 steps, which covers the relative refractory period (10 ms) when the stimulation frequency is less than 100Hz. We have now included this number in the Methods.

2.29) Section: Greedy temporal dithering outperforms open loop methods.Regarding the number of single-electrode stimuli used in image reconstruction, it would be better to place the numbers used in the context of what is possible in the perceptual time window. It would recommend using the value of 333 instead of 500, as this corresponds to the number of 0.15 pulses that could be fit into a 50 ms window. The value of 3000 roughly corresponds to what might be achieved with spatial multiplexing. The value of 10,000 corresponds to the upper limit that is achievable through this algorithm.

This is an important and somewhat subtle point. While these calculations are correct for the primate recording, we found that 225±29 electrical stimulations were needed for asymptotic reconstruction performance of checkerboard targets in the rat retina. The total number of electrical stimulations needed depends on many factors including the number of cells targeted, their expected firing rates for the visual image, and the distribution of RGC activation probabilities in the electrical stimulation dictionary. Future work will be needed to identify how these factors vary across individuals, species and neural circuits. We have added text in the Discussion highlighting this issue.

2.30) I think it would be beneficial to make it clearer that the results in Figure 3 are simulated. It would also strengthen the study to perform validation in ex vivo retina to apply the greedy temporal dithering stimuli to the retina and reconstruct the image from the responses. If there is a good reason not to do this, this should be explained.

We agree that direct closed-loop experimental validation with the temporally dithered stimulation is important. We have now performed these experiments and clarified our language throughout the manuscript. Please see our responses to Essential Revisions 1 and 3.

2.31) It would improve the study if a reconstruction algorithm that provides an image with a better match to the perception of phosphenes by retinal implant users was used. If this cannot be done, it should be discussed as a limitation of the study.

This would be a highly relevant point if our electrical stimulation approaches had the same coarse level of control as existing implants. But in fact, the situation is quite different in the present work. For the short (150 µs) and low current (<4 µA) pulses we use, only single or small groups of cells tend to be electrically activated (Figure 1C) (Sekirnjak et al., 2006, 2008). This is in part because the mechanism of activation is direct depolarization, rather than a network-mediated excitation, and in part because we explicitly avoid activation of axons (which produces large phosphenes in existing implants (Beyeler et al., 2019)) by pre-calibration. Due to these fundamental differences in stimulation compared to existing retinal implants, we do not expect the perception of phosphenes of the kind seen in present-day implants. We have clarified these considerations in the Results and Discussion.

2.32) It would be helpful to clarify some details of the implementation of the open-loop strategy including:How the area that was "near" the electrode was selected when calculating the intensity of the visual stimulus.How the temporal sequence of the electrodes was chosen.

We have replaced “open-loop” with “static pixel-wise mapping” to more accurately reflect the calculation we are performing which approximates the function of present-day retinal implants. Specifically, we used a mapping between the intensity of the visual stimulus incident on an electrode and the intensity of the current passed through that electrode to determine which electrical stimuli to deliver. The intention of this analysis is to provide a generous benchmark for present-day devices – it is generous in the sense that we assume a much greater degree of calibration precision than these devices can actually achieve.

First, an affine transformation mapped the visual stimulus onto the electrode array. Second, the average visual stimulus intensity was identified over an approximately 130 µm x 130 µm region around the electrode location. Third, the average visual stimulus intensity on the electrode (*s*) was mapped to the current amplitude (*i*) using a scaled sigmoid:

i=a+b/(1+exp⁡(cs+d)) This electrical stimulus was then delivered *n* times at that electrode. All parameters (i.e. five parameters {*a*, *b*, *c*, *d*, *n*} for each electrode) were simultaneously optimized to minimize reconstruction error across a training set of random checkerboard images.

A temporal dithering strategy for ordering the electrical stimuli was assumed so that the stimuli would not interfere with one another. The performance of this static pixel-wise mapping approach was evaluated in the same way as the dynamically optimized stimulation approach. In particular, the target visual stimulus was linearly reconstructed from the identified electrical stimulation sequence using samples drawn from the single-electrode calibration data.

The static mapping at each electrode captures the common aspect of existing approaches and highlights the crucial improvement of our dynamic approach. Again, we note that this procedure provides a generous interpretation of existing methods because it uses actual measured neural responses for optimizing the mapping rather than relying on more limited patient feedback. We have updated the text in the Results and clarified these implementation details in the Methods.

2.33) Section: Greedy temporal dithering is nearly optimal given the interface constraints.A realistic number of stimulation pulses should be used in performing this comparison e.g. < 400 for the pure temporal dithering or < 3000 for the spatially multiplexed, temporal dithering.

Please see our response to Recommendations for Authors 2.29 above.

2.34) It would be helpful to clarify some details of the implementation of the open-loop strategy including:The meaning and purpose of the term V^T.w in the objective function.Whether w>=0 was required after the integer requirement was relaxed in the optimization.

These details of the implementation of the optimal comparison strategy have been clarified in the Methods. Yes, the non-negativity constraint keeps the approximate objective closer to the original formulation.

2.35) Section: Spatial multiplexing for fitting multiple stimuli in a visual integration window.As described in my public review, the description of independence is not correct. I have suggested an alternative description that I believe accords with what was actually implemented.

Thank you for pointing out the inconsistency in the description of spatial independence. We have corrected and simplified the description in the Methods.

2.36) It was surprising that the results of the validation experiments on ex vivo retina with the spatially multiplexed, temporally dithered algorithm were not reported more thoroughly. It is important to provide figures showing the image reconstruction that was achieved and the statistics for the reconstruction error.

We have now performed the closed-loop validation experiment, and added reconstructed images from it. Please see our responses to Essential Revisions 1 above and the changes to the manuscript.

Reviewer #33.1) In this study, Shah and colleagues propose an interesting solution to the non-linear interactions caused by simultaneously stimulating multiple electrodes within a retinal implant. Through high-resolution recordings of ON and OFF parasol retinal ganglion cells, the authors demonstrate that a greedy dithering and spatially multiplexed algorithm, which can also work in the presence of saccadic eye movements, is able to faithfully reconstruct images represented by total numbers of spikes in a given time window across multiple retinal ganglion cells. Essentially, Shah and colleagues propose and demonstrate a method to only stimulate single or groups of 8 electrodes at a time from a pre-established dictionary, but then interleave stimulation of multiple electrodes or groups rapidly across the dictionary to additively build an image. Through their very rigorous and elegant ex vivo recordings in 180 ON and OFF parasol cells across four primate retina preparations, the authors compellingly demonstrate that (i) their greedy algorithm performs better than an open loop algorithm, similar to an optimal algorithm considering the interface constraints, and close but not equal to an ideal control using only a single-electrode dictionary; (ii) that groups of electrodes can be simultaneously activated with a high-resolution neural interface without any retinal interactions provided that they are at least 160 μm apart; (iii) that the algorithm performs just as well even with only 50% of the electrodes on the interface and (iv) that the algorithm can work in the presence of saccadic eye movements and performs better when both saccadic and fixational eye movements are made as opposed to saccadic movements alone.The experimental recordings and performance of the algorithm in various conditions are the biggest strengths of this study and the authors certainly demonstrate that their algorithm can reproduce spiking numbers across an array of cells that resemble closely spiking numbers evoked by visual stimuli for these conditions. In other words, the authors' primary claim that the neural code for visual images in the retina (in the form of spiking numbers) can be faithfully reproduced with electrical stimulation using such an algorithm, is well supported by evidence.A major weakness in the study however is the reliance of this algorithm on several significant assumptions about neural coding in the retina, neural coding in the visual brain, and interactions between electrodes even with non-simultaneous stimulation. Some of these assumptions have already been highly challenged in several studies in the visual neuroscience field and in studies involving the perception of phosphenes with interleaved stimulation of single electrodes.Therefore, in light of what is currently known about visual encoding and artificial vision, the study whilst showcasing an elegant computational tool perhaps provides only little hope that such an algorithm will actually work in practice to recreate the perception of images with electrical stimulation but instead does lay a foundation for further work to be done with the assessment of future algorithms. The main assumptions that the authors rely on include:1) That neural coding in the retina is simply based on a number of spikes evoked by populations of cells ignoring any temporal patterns of responses. A plethora of studies has indicated that relative spike timing between groups of retinal ganglion cells for example can encode complex visual features but the greedy algorithm does not aim to mimic these spike timing features.

Please see our response to Recommendations for Authors 3.2 below.

2) That perception within the brain is solely based on a number of spikes within a slow temporal integration window (the authors cite a 1995 reference for this). Since 1995 though, this has also been challenged, therefore extending the authors' claims of reproducing spike numbers in the retina to reproducing perception in the brain would be contentious.

Please see our response to Essential Revisions 2 paragraph 3 above.

3) That neural interactions with non-simultaneous interleaved electrical stimulation are absent. There is in silico, electrophysiological and perceptual evidence with retinal implants that interleaving of electrodes still results in neural interactions and that perception with interleaved stimulation with multiple electrodes does not result in a linear summed perception of phosphenes evoked by single electrodes i.e. dictionary elements. Therefore, the algorithm would only work if such interactions are minimal or absent, for example with larger than 0.1 ms intervals between stimulations or more than 160 μm electrode separation. Note, interactions with interleaving also exist with cochlear implants as the current spread is large.

Please see our response to Essential Revisions 2 paragraph 4 above.

4) That even if the above 3 assumptions were applied and true, the algorithm can faithfully extrapolate to reconstruct moving images at 24 per second. This seems unlikely as presumably the total time required to linearly reconstruct a single static image would extend to many tens or even hundreds of ms given the number of times each dictionary element needs to be accessed to enable reproduction of similar spiking numbers between visual and electrical stimulation, runs in the thousands.

Please see our response to Recommendations for Authors 3.3 below.

In spite of major reliance on these assumptions, the authors do demonstrate a very useful tool in the form of the greedy algorithm for situations perhaps other than the visual system, where perception with artificial stimulation may be more predictable and interactions with non-simultaneous stimulation may be simpler.

Thank you for recognizing the strength of the work.

Recommendations for the authors:3.2) It may be possible to address at least some of the limitations in particular (1) and (4) mentioned in the public review. For limitation (1), the authors could try and experiment with their algorithm and reanalyse data to examine if and how well spike timing features (perhaps relative first spike latencies between RGCs or other temporal patterns of spikes) are reproducible.

This is an important point. Please see our response to Essential Revisions 2 paragraph 2. We also note that with the very high temporal precision of our stimulation (evoked spike time variation of roughly 0.1 ms), if there were some degree of stimulus coding in the relative timing of spikes, that relative timing could certainly be reproduced by the stimulation sequences that we provide, with a suitable modification of the optimization approach. However, this would substantially increase the overall complexity of the algorithm and we think it is beyond the scope of this paper.

3.3) For limitation (4) the authors could at least perform calculations of time taken by the algorithm in each of the situations and targets presented, to examine if these times are realistic.

In fact, realistic numbers of electrical stimulations were required for the closed-loop experimental validation of greedy temporal dithering using rat retina (225±29 stimulations for asymptotic reconstruction performance, requiring 62±9 ms to deliver the temporally dithered sequence). For the thousands of stimulations reported for the macaque retina, spatial multiplexing could reduce the delivery time down to a realistic tens of milliseconds. For example, delivering 3000 stimuli at a 0.15 ms interval would require 450 ms without spatial multiplexing, but could require as short as 57 ms with spatial multiplexing delivering an average of 8 electrical stimuli per time step. While the former duration exceeds visual integration time, the latter duration approximately matches it. The exact number of electrical stimulations needed for each situation depends on many factors including the number of cells targeted, their expected firing rates for the particular visual image, and the distribution of RGC activation probabilities in the electrical stimulation dictionary. Future work will be needed to identify how these factors vary across individuals, species and neural circuits. We have added text in the Discussion highlighting this issue.

3.4) For limitations (2) and (3), the authors at a minimum should address these clearly in their discussion and the potential implications of the failure of these assumptions on their algorithm performance.

Please see our response to Essential Revisions 2 paragraphs 3 and 4 for a discussion of limitations (2) and (3), respectively. We have clarified these considerations in the Discussion.

3.5) Other things that the authors should consider is including some example raw data from their retinas before and after artifact subtraction in response to both visual targets and their greedy algorithm as a figure.

Thank you for this suggestion. While this could be beneficial, we ultimately decided against this as the spike sorting in the presence of electrical artifact is a very involved topic and has been extensively covered in other papers from our group (Gogliettino et al., 2023; Jepson et al., 2013, 2014; Madugula et al., 2022; Sekirnjak et al., 2006, 2008). We cite some of these papers in the Methods section.

**References**

Alonso, J. M., Usrey, W. M., & Reid, R. C. (1996). Precisely correlated firing in cells of the lateral geniculate nucleus. *Nature*, *383*(6603), 815–819. https://doi.org/10.1038/383815a0

Berry, M. J., & Meister, M. (1998). Refractoriness and Neural Precision. *Journal of Neuroscience*, *18*(6), 2200–2211.https://doi.org/10.1523/JNEUROSCI.18-06-02200.1998

Berry, M. J., Warland, D. K., & Meister, M. (1997). The structure and precision of retinal spike trains. *Proceedings of the National Academy of Sciences*, *94*(10), 5411–5416. https://doi.org/10.1073/pnas.94.10.5411

Beyeler, M., Nanduri, D., Weiland, J. D., Rokem, A., Boynton, G. M., & Fine, I. (2019). A model of ganglion axon pathways accounts for percepts elicited by retinal implants. *Scientific Reports*, *9*(1), 9199. https://doi.org/10.1038/s41598-019-45416-4

Borghuis, B. G., Tadin, D., Lankheet, M. J. M., Lappin, J. S., & van de Grind, W. A. (2019). Temporal Limits of Visual Motion Processing: Psychophysics and Neurophysiology. *Vision*, *3*(1), 5. https://doi.org/10.3390/vision3010005

Brackbill, N., Rhoades, C., Kling, A., Shah, N. P., Sher, A., Litke, A. M., & Chichilnisky, E. J. (2020). Reconstruction of natural images from responses of primate retinal ganglion cells. *eLife*, *9*, e58516. https://doi.org/10.7554/*eLife*.58516

Chichilnisky, E. J., & Kalmar, R. S. (2003). Temporal Resolution of Ensemble Visual Motion Signals in Primate Retina. *The Journal of Neuroscience*, *23*(17), 6681–6689. https://doi.org/10.1523/JNEUROSCI.23-17-06681.2003

Choi, J. S., Brockmeier, A. J., McNiel, D. B., Kraus, L. M. von, Príncipe, J. C., & Francis, J. T. (2016). Eliciting naturalistic cortical responses with a sensory prosthesis via optimized microstimulation. *Journal of Neural Engineering*, *13*(5), 056007. https://doi.org/10.1088/1741-2560/13/5/056007

Frechette, E. S., Sher, A., Grivich, M. I., Petrusca, D., Litke, A. M., & Chichilnisky, E. J. (2005).

Fidelity of the Ensemble Code for Visual Motion in Primate Retina. *Journal of Neurophysiology*, *94*(1), 119–135. https://doi.org/10.1152/jn.01175.2004

Gogliettino, A. R., Madugula, S. S., Grosberg, L. E., Vilkhu, R. S., Brown, J., Nguyen, H., Kling, A., Hottowy, P., Dąbrowski, W., Sher, A., Litke, A. M., & Chichilnisky, E. J. (2023).

High-Fidelity Reproduction of Visual Signals by Electrical Stimulation in the Central Primate Retina. *Journal of Neuroscience*, *43*(25), 4625–4641. https://doi.org/10.1523/JNEUROSCI.1091-22.2023

Gollisch, T., & Meister, M. (2008). Rapid Neural Coding in the Retina with Relative Spike Latencies. *Science*, *319*(5866), 1108–1111. https://doi.org/10.1126/science.1149639

Gütig, R., Gollisch, T., Sompolinsky, H., & Meister, M. (2013). Computing Complex Visual Features with Retinal Spike Times. *PLoS ONE*, *8*(1), e53063. https://doi.org/10.1371/journal.pone.0053063

Haji Ghaffari, D., Akwaboah, A. D., Mirzakhalili, E., & Weiland, J. D. (2021). Real-Time Optimization of Retinal Ganglion Cell Spatial Activity in Response to Epiretinal Stimulation. *IEEE Transactions on Neural Systems and Rehabilitation Engineering*, *29*, 2733–2741. https://doi.org/10.1109/TNSRE.2021.3138297

Ho, E., Shmakov, A., & Palanker, D. (2020). Decoding network-mediated retinal response to electrical stimulation: Implications for fidelity of prosthetic vision. *Journal of Neural Engineering*, *17*(6), 10.1088/1741-2552/abc535. https://doi.org/10.1088/1741-2552/abc535

Jepson, L. H., Hottowy, P., Mathieson, K., Gunning, D. E., Dabrowski, W., Litke, A. M., & Chichilnisky, E. J. (2013). Focal electrical stimulation of major ganglion cell types in the primate retina for the design of visual prostheses. *The Journal of Neuroscience: The Official Journal of the Society for Neuroscience*, *33*(17), 7194–7205. https://doi.org/10.1523/JNEUROSCI.4967-12.2013

Jepson, L. H., Hottowy, P., Mathieson, K., Gunning, D. E., Dąbrowski, W., Litke, A. M., & Chichilnisky, E. J. (2014). Spatially patterned electrical stimulation to enhance resolution of retinal prostheses. *The Journal of Neuroscience: The Official Journal of the Society for Neuroscience*, *34*(14), 4871–4881. https://doi.org/10.1523/JNEUROSCI.2882-13.2014

Keat, J., Reinagel, P., Reid, R. C., & Meister, M. (2001). Predicting Every Spike: A Model for the Responses of Visual Neurons. *Neuron*, *30*(3), 803–817. https://doi.org/10.1016/S0896-6273(01)00322-1

Li, P. H., Gauthier, J. L., Schiff, M., Sher, A., Ahn, D., Field, G. D., Greschner, M., Callaway, E. M., Litke, A. M., & Chichilnisky, E. J. (2015). Anatomical identification of extracellularly recorded cells in large-scale multielectrode recordings. *The Journal of Neuroscience: The Official Journal of the Society for Neuroscience*, *35*(11), 4663–4675.https://doi.org/10.1523/JNEUROSCI.3675-14.2015

Lotlikar, A., Shah, N. P., Gogliettino, A. R., Vilkhu, R., Madugula, S., Grosberg, L., Hottowy, P., Sher, A., Litke, A., Chichilnisky, E. J., & Mitra, S. (2023). Partitioned Temporal Dithering for Efficient Epiretinal Electrical Stimulation. *2023 11th International IEEE/EMBS Conference on Neural Engineering (NER)*, 1–5. https://doi.org/10.1109/NER52421.2023.10123787

Madugula, S. S., Gogliettino, A. R., Zaidi, M., Aggarwal, G., Kling, A., Shah, N. P., Brown, J. B.,

Vilkhu, R., Hays, M. R., Nguyen, H., Fan, V., Wu, E. G., Hottowy, P., Sher, A., Litke, A. M., Silva, R. A., & Chichilnisky, E. J. (2022). Focal Electrical Stimulation of Human Retinal Ganglion Cells for Vision Restoration. *Journal of Neural Engineering*, *19*(6), 10.1088/1741-2552/aca5b5. https://doi.org/10.1088/1741-2552/aca5b5

Meister, M. (1996). Multineuronal codes in retinal signaling. *Proceedings of the National Academy of Sciences*, *93*(2), 609–614. https://doi.org/10.1073/pnas.93.2.609

Reich, D. S., Victor, J. D., Knight, B. W., Ozaki, T., & Kaplan, E. (1997). Response Variability and Timing Precision of Neuronal Spike Trains in vivo. *Journal of Neurophysiology*, *77*(5), 2836–2841. https://doi.org/10.1152/jn.1997.77.5.2836

Samaha, J., & Postle, B. R. (2015). The speed of α-band oscillations predicts the temporal resolution of visual perception. *Current Biology : CB*, *25*(22), 2985–2990. https://doi.org/10.1016/j.cub.2015.10.007

Sekhar, S., Ramesh, P., Bassetto, G., Zrenner, E., Macke, J. H., & Rathbun, D. L. (2020).Characterizing Retinal Ganglion Cell Responses to Electrical Stimulation Using Generalized Linear Models. *Frontiers in Neuroscience*, *14*, 378. https://doi.org/10.3389/fnins.2020.00378

Sekirnjak, C., Hottowy, P., Sher, A., Dabrowski, W., Litke, A. M., & Chichilnisky, E. J. (2006). Electrical stimulation of mammalian retinal ganglion cells with multielectrode arrays. *Journal of Neurophysiology*, *95*(6), 3311–3327. https://doi.org/10.1152/jn.01168.2005

Sekirnjak, C., Hottowy, P., Sher, A., Dabrowski, W., Litke, A. M., & Chichilnisky, E. J. (2008). High-resolution electrical stimulation of primate retina for epiretinal implant design. *The Journal of Neuroscience: The Official Journal of the Society for Neuroscience*, *28*(17), 4446–4456. https://doi.org/10.1523/JNEUROSCI.5138-07.2008

Shadlen, M. N., & Newsome, W. T. (1994). Noise, neural codes and cortical organization. *Current Opinion in Neurobiology*, *4*(4), 569–579. https://doi.org/10.1016/0959-4388(94)90059-0

Tadin, D., Lappin, J. S., Blake, R., & Glasser, D. M. (2010). High temporal precision for perceiving event offsets. *Vision Research*, *50*(19), 1966–1971. https://doi.org/10.1016/j.visres.2010.07.005

Tafazoli, S., MacDowell, C. J., Che, Z., Letai, K. C., Steinhardt, C. R., & Buschman, T. J. (2020). Learning to control the brain through adaptive closed-loop patterned stimulation. *Journal of Neural Engineering*, *17*(5), 056007. https://doi.org/10.1088/1741-2552/abb860

Uzzell, V. J., & Chichilnisky, E. J. (2004). Precision of Spike Trains in Primate Retinal Ganglion Cells. *Journal of Neurophysiology*, *92*(2), 780–789. https://doi.org/10.1152/jn.01171.2003

Vasireddy, P. K., Gogliettino, A. R., Brown, J. B., Vilkhu, R. S., Madugula, S. S., Phillips, A. J., Mitral, S., Hottowy, P., Sher, A., Litke, A., Shah, N. P., & Chichilnisky, E. J. (2023).Efficient Modeling and Calibration of Multi-Electrode Stimuli for Epiretinal Implants. *2023 11th International IEEE/EMBS Conference on Neural Engineering (NER)*, 1–4. https://doi.org/10.1109/NER52421.2023.10123907

Wu, E. G., Brackbill, N., Rhoades, C., Kling, A., Gogliettino, A. R., Shah, N. P., Sher, A., Litke, A. M., Simoncelli, E. P., & Chichilnisky, E. J. (2023). *Fixational Eye Movements Enhance the Precision of Visual Information Transmitted by the Primate Retina* (p.2023.08.12.552902). bioRxiv. https://doi.org/10.1101/2023.08.12.552902

Wutz, A., Muschter, E., van Koningsbruggen, M. G., Weisz, N., & Melcher, D. (2016). Temporal Integration Windows in Neural Processing and Perception Aligned to Saccadic Eye Movements. *Current Biology: CB*, *26*(13), 1659–1668. https://doi.org/10.1016/j.cub.2016.04.070

Yoon, Y. J., Lee, J.-I., Jang, Y. J., An, S., Kim, J. H., Fried, S. I., & Im, M. (2020). Retinal Degeneration Reduces Consistency of Network-mediated Responses Arising in Ganglion Cells to Electric Stimulation. *IEEE Transactions on Neural Systems and Rehabilitation Engineering : A Publication of the IEEE Engineering in Medicine and Biology Society*, *28*(9), 1921–1930. https://doi.org/10.1109/TNSRE.2020.3003345

Zaidi, M., Aggarwal, G., Shah, N. P., Karniol-Tambour, O., Goetz, G., Madugula, S. S., Gogliettino, A. R., Wu, E. G., Kling, A., Brackbill, N., Sher, A., Litke, A. M., & Chichilnisky, E. J. (2023). Inferring light responses of primate retinal ganglion cells using intrinsic electrical signatures. *Journal of Neural Engineering*, *20*(4), 045001. https://doi.org/10.1088/1741-2552/ace657